# Evolution of tissue and developmental specificity of transcription start sites in Bos taurus indicus

Mehrnush Forutan [1✉], Elizabeth Ross[1], Amanda J. Chamberlain [2], Loan Nguyen[1], Brett Mason[2], Stephen Moore[1], Josie B. Garner[3], Ruidong Xiang[4] & Ben J. Hayes [1]

To further the understanding of the evolution of transcriptional regulation, we profiled genome-wide transcriptional start sites (**TSSs**) in two sub-species, *Bos taurus taurus* and *Bos taurus indicus*, that diverged approximately 500,000 years ago. Evolutionary and developmental-stage differences in TSSs were detected across the sub-species, including translocation of dominant TSS and changes in TSS distribution. The 16% of all SNPs located in significant differentially used TSS clusters across sub-species had significant shifts in allele frequency (472 SNPs), indicating they may have been subject to selection. In spleen and muscle, a higher relative TSS expression was observed in *Bos indicus* than *Bos taurus* for all heat shock protein genes, which may be responsible for the tropical adaptation of *Bos indicus*.

[1] Centre for Animal Science, Queensland Alliance for Agriculture and Food Innovation, The University of Queensland, Brisbane, QLD, Australia. [2] Agriculture Victoria, Centre for AgriBiosciences, Bundoora, VIC, Australia. [3] Agriculture Victoria, Animal Production Sciences, Ellinbank Dairy Centre, Ellinbank, VIC, Australia. [4] Faculty of Veterinary & Agricultural Science, The University of Melbourne, Parkville, VIC, Australia. ✉email: m.forutan@uq.edu.au

Understanding how the expression of genes is regulated is an essential goal of genomics. Gene regulation mechanisms contribute to evolutionary processes associated with species divergence, and within species divergence such as breed formation[1]. Variations in gene expression is largely due to cis mechanisms where regulatory molecules bind to elements such as promoters and enhancers close to genes to initiate transcription. Transcription Start Sites (TSSs) act as an integration region for a wide range of molecular signals to control transcription and expression levels[2–5]. Previous studies[6,7], assumed that promoters have a TATA-box, which directs the positioning of the pre-initiation complex, in effect initiating transcription from a single nucleotide. In contrast, more recent studies[8] have shown that the majority of human and mouse RNA Polymerase II core promoters have an array of closely positioned TSSs instead of the expected single TSS. In agreement with this finding, the FANTOM consortium project highlighted that few genes are true 'housekeeping' (considered to be genes with one TSS in some definitions), whereas many mammalian promoters are composed of several adjacent TSSs[9]. Furthermore, a large number of genes have several strong core promoters, which force alternative splicing and ultimately, production of different protein isoforms[8,10]. There is evidence in human genes that different isoforms are produced as a result of the usage of alternative TSSs[11,12]. These results led to the "adaptive hypothesis"; that alternative TSSs are a widely used, regulated mechanism to expand the transcriptome diversity[9]. However, the vast majority of TSSs have unknown functions. For example more than 90,000 TSSs are annotated for ~20,000 human protein-coding genes in the ENSEMBL genome reference consortium human build 37[13]. Accordingly, a recent study proposed an alternative hypothesis that there is only one optimal TSS per gene and that other TSSs arise from errors in transcriptional initiation sites[13].

Several efficient technologies have been developed to dissect gene regulation mechanisms recently, including ChIP-seq[14] and ATAC-seq[15]. These technologies detect sites across the genome, which can be used to infer regulatory elements including promoters and enhancers. These technologies are also used as the primary tools for consortia to annotate human (ENCODE[16]) and animal (FAANG[17]) genomes. The FANTOM consortium has focused on the mapping of TSSs using Cap Analysis of Gene Expression (CAGE) to identify promoters and enhancers across a large collection of primary cell types in the human and mouse genomes[9,18].

CAGE has been developed as one of the main high-throughput assays for studying TSSs and their expression[19]. Sequencing short reads (or tags) from the 5'end of full-length cDNA allows TSSs to be mapped and their expression to be studied. A specific advantage of the CAGE method is that reads mapped to the genome provide accurate location of TSS and quantify transcription[8,20]. As CAGE tags can be aligned to a reference genome without the need for transcript annotations, it can detect not only TSSs of known mRNAs but also mRNA from alternative TSSs that might often be tissue or developmental-stage specific[21]. In animals, a CAGE and TSS based genome annotation database has being built in sheep[22]. In cattle, TSSs have been investigated using RNA Annotation and Mapping of Promoters for the Analysis of Gene Expression (RAMPAGE)[23]. However, to our knowledge, there is no study that examined the tissue, developmental specificity of TSSs using CAGE, and the regulatory mechanisms centred around TSS which contribute to the differentiation of the two cattle subspecies, Bos taurus taurus and Bos taurus indicus, which diverged up to 0.5 million years ago.

It has been well established that changes in transcriptional regulation underlie much of the phenotypic variation between species[24], and there is some evidence for gene expression divergence between even closely related lineages (e.g., ref.[25]). Previous research has shown that transcription factor binding sites[26], centromeres[27], and TSSs[8] are affected by gain-and-loss of functional genetic elements, called "turnover". Thus far, TSS locations have mostly been explored and compared between human and mouse (e.g., ref.[28,29]), which diverged approximately 96 million years ago[30]. In this study we exploit a much closer evolutionary split between Bos taurus taurus and Bos taurus indicus using CAGE datasets for the first time, to the best of our knowledge, to assess changes in TSSs for closely related (cattle) species, with the aim of gaining further insights into the evolution of TSSs. CAGE-Seq (CAGE followed by sequencing) was performed on 11 tissues at adult stages, including liver, lung, kidney, thyroid, spleen, muscle, uterus, ovary, blood in indicus and liver, spleen, muscle, mammary, heart in taurus subspecies, and two tissues in fetal stage, including liver and lung in indicus and liver in taurus subspecies. To the best of our knowledge, this paper highlights evolutionary divergence in TSS usage and is the first bovine TSS discovery study using CAGE-Seq data.

## Results and discussion

**Evolutionary divergence in TSS usage comparing Bos taurus and Bos indicus.** We first conducted analyses to understand the impact of sequence coverage on our results. Descriptive analysis of number of reads in bam files after quality control, CAGE tags starting sites (CTSSs) and TSSs indicated a positive correlation of 0.98 (0.62) and 0.92 (0.72) between number of reads in bam files and the number of CTSSs (TSSs) detected in Bos indicus and Bos taurus subspecies, respectively. An analysis of the technical reproducibility of TSSs from CAGE samples with different coverage (total coverage or in silico produced half coverage) were investigated for TSS calling. For this purpose, fetal lung, adult liver, and lung, respectively with low, medium and high sample coverage were randomly divided into two subsamples and TSSs called in each. Technical Reproducibility was measured as the fraction of consensus TSS clusters (see Methods) commonly observed in the total and half samples. A minimum technical reproducibility of 73% was obtained when the total sample coverage was less than 0.08 million CTSSs supported by 3 or more CAGE read 5'-ends for Bos indicus fetal lung and Bos taurus adult liver samples (Supplementary Tables 1-2) The maximum technical reproducibility of 99% between total and half samples was observed in adult Bos indicus lung and liver tissues with more than 0.14 million CTSSs supported by 3 or more CAGE read 5'-ends in the total sample. To further investigate the effect of coverage on TSS diversity of each consensus TSS cluster we measured Pearson correlation between Shannon index of TSS diversity of consensus TSS clusters in the half and total samples (Supplementary Fig. 1A–E). Consistent with above results, the highest (0.96) and lowest (0.80) correlation was achieved in adult and fetal Bos indicus lung, respectively (Supplementary Fig. 1D-E). Although the reproducibility of TSSs was principally a consequence of the depth of sampling, it was quite similar in Bos indicus adult lung and liver tissues having about 0.56 and 0.14 million CTSSs supported by 3 or more CAGE tags (Supplementary Fig. 1C-D). For most of our samples, we had more than 0.1 million CTSSs supported by 3 or more CAGE read 5'-ends (Supplementary Table 1), reflecting that our libraries have been sequenced deep enough.

To study evolutionary TSS changes, TSSs from both subspecies were first aggregated into a single set of consensus TSS clusters, for each tissue (liver, spleen, and muscle) (Supplementary Data 1–3). Then consensus TSS clusters within each tissue were used for detecting divergent TSS clusters (clusters only observed in one subspecies), comparing dominant TSS positions (CTSS with highest number of CAGE tags in the TSS cluster), identifying

differentially used TSS clusters, and evaluating the diversity of the tags within TSS clusters across three tissues. The workflow illustrating the entire flow of the analysis is shown in Fig. 1. To support our conclusion from comparative analysis between subspecies, and to assign the reported differences to the subspecies effect, the same workflow was applied to the available *taurus* spleen, muscle, and liver biological replicates. This resulted in an estimation of the expected differences observed between samples from the same subspecies. The number of consensus TSSs between *taurus* and *indicus* subspecies and between *taurus* biological replicates are shown in Table 1 and Supplementary Table 3, respectively. In agreement with a previous study[31] our results showed that a large majority of TSSs clusters had only a very small number of tags, were observed in only a single sample and were lowly expressed. By increasing the number of CAGE-Seq reads mapped to the TSS clusters (**TPM**), the proportion of divergent TSS clusters was reduced (Figs. 2–4A; Supplementary Figs. 2–4A). In total, 1%, 2%, and 14% of the consensus clusters with at least 10 TPM expression in spleen, muscle, and liver were observed only in one subspecies, respectively (Figs. 2–4A). However, the proportion of divergent consensus TSSs between *taurus* replicates for consensus TSSs with at least 10 TPM expression in spleen, muscle and liver was 0.5%, 0.2%, and 0.5%, respectively. A significantly higher proportion of divergent TSSs was observed across subspecies compared to within *Bos taurus* subspecies in spleen and liver (Bootstrap-based *P*-value < 0.05).

To evaluate the diversity of the tags within TSS clusters, TSS diversity was analysed using the Shannon index[32] for each consensus cluster, and the correlation between subspecies and within *Bos taurus* subspecies across clusters were compared for each tissue (Figs. 2–4B; Supplementary Figs. 2–4B). Shannon index is commonly used in biodiversity research and in an analysis of TSS clusters. The Shannon index will rise with the number of TSSs in a cluster as well as with the evenness of the relative uses of these TSSs, and conversely will be zero if one dominant TSS is used[32]. The TSS diversity only had higher correlation between *taurus* biological replicates compared to between subspecies groups in liver tissue (0.86% vs. 0.51%). However, in spleen and muscle the TSS diversity were relatively more correlated between subspecies compared to between biological replicates. The lower correlation of TSS diversity between *taurus* replicates compared to between subspecies groups for spleen and muscle could reflect inadequate sample coverage of individual biological replicates. We tried to compensate for this by merging them together and creating a union of CTSSs present in individual replicates and raw tag counts for those CTSSs in two replicates when comparing TSSs between subspecies group.

Because TSSs are less reproducible when the number of sequencing reads mapped to a gene is too small, only consensus TSSs with at least 10 TPM reads in a sample were included in the comparison of dominant TSS position. Dominant TSSs were

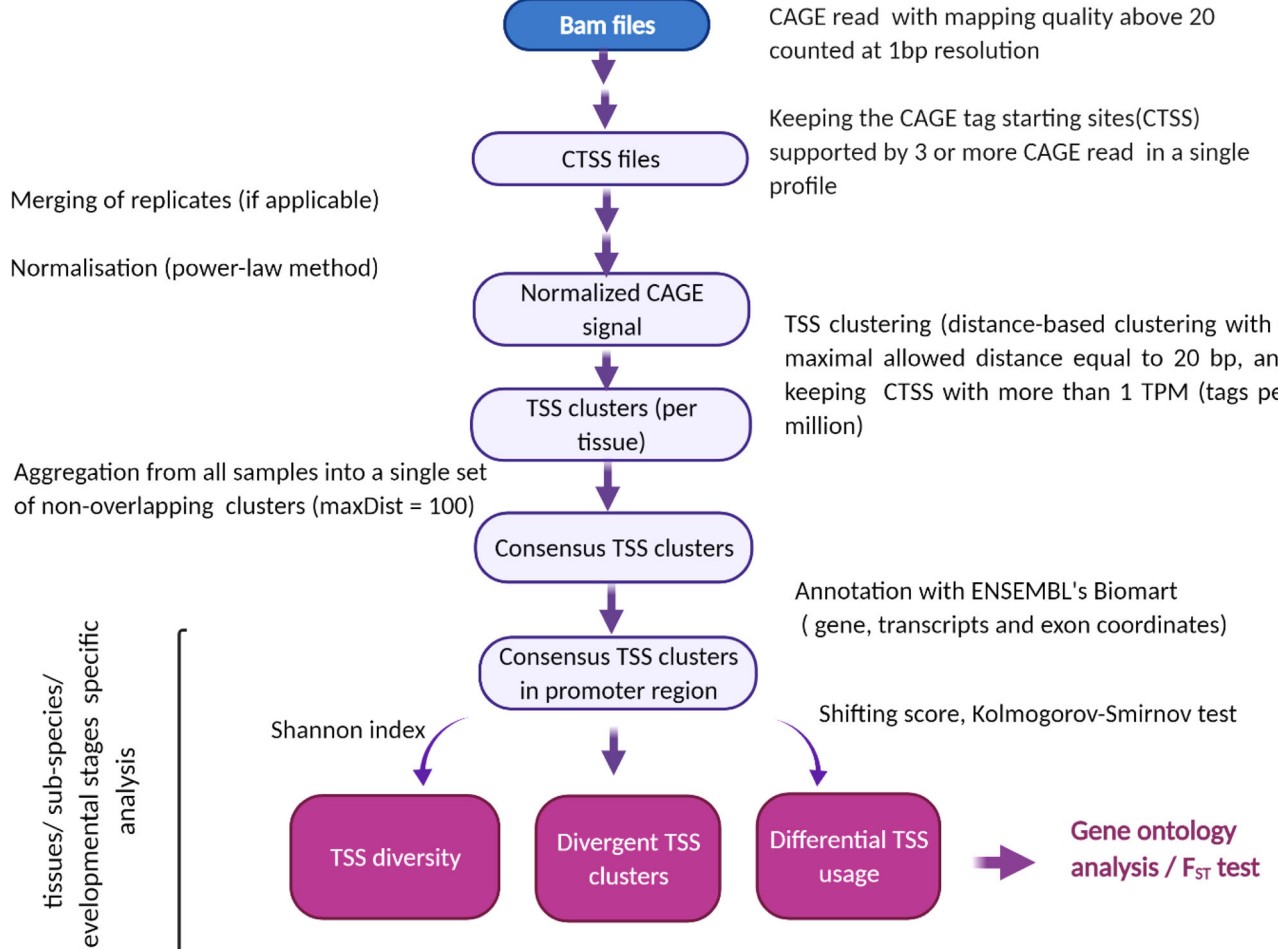

**Fig. 1 Flow chart of main steps in TSS identification.** CAGE-seq signals were normalized based on power-law method after doing some quality control for bam and CTSS files. Then TSSs were called using distance-based clustering methods and aggregated into a consensus TSS clusters. Eventually, different properties of TSSs were assessed after annotating consensus TSS cluster with ENSEMBL.

**Table 1 Evolutionary comparison of consensus TSS clusters between *Bos taurus* and *Bos indicus* subspecies in three adult tissues (spleen, liver, and muscle).**

| Tissue | Number of consensus TSS clusters across subspecies | | Number of consensus TSS clusters with at least 10 TPM expression level | | Number of divergent TSSs[a] | Number of significant TSSs with differential usage[a] and shifting score >0.1 | |
|---|---|---|---|---|---|---|---|
| | Total number | In promoter regions | Total number | In promoter regions | | Total number | Significant (P-value < 0.05) |
| Spleen | 17,361 | 10,844 | 7,254 | 5,795 | 50 | 1,174 | 71 |
| Liver | 16,255 | 10,684 | 7,112 | 5,622 | 799 | 1,671 | 600 |
| Muscle | 12,502 | 8,118 | 4,596 | 3,613 | 68 | 519 | 48 |

aWith at least 10 TPM expression level in promoter region.

mostly located in the same position between subspecies samples (75%, 74%, 55%) and between *taurus* replicates (72%, 79%, 78%) in spleen, muscle, and liver, respectively (Figs. 2–4C; Supplementary Figs. 2–4C). Only in liver was the proportion of dominant TSS translocations significantly higher across subspecies than between *Bos taurus* biological replicates (Bootstrap-based *P*-value < 0.05). Our results confirmed that dominant TSS positions on the genome were not fixed between/within subspecies. This is consistent with previous evolutionary research between human and mouse that suggested that the TSS locations are highly flexible and evolvable[28]. A previous study analysing the different causes of alternative splicing events in cattle confirmed that the high percentage (69% and 83%) of bovine gene set affected by splicing had alternative TSSs and termination sites, respectively[33].

The position and the fraction of individual TSSs used within one consensus TSS cluster can be substantially different between subspecies group. To detect differentially used TSS clusters, the shifting score, the degree of physical separation of TSSs, and statistical significance of differential TSSs usage using Kolmogorov–Smirnov test was calculated for all consensus clusters between the two subspecies for each tissue in CAGEr[34] (see Methods). About 20% (28%), 14% (30%), 36% (18%) of consensus TSSs with at least 10 TPM expression had at least 10% of the transcription in one subspecies (biological replicate), which was independent and located outside of the region used to initiate transcription from the same TSS cluster in the other subspecies (biological replicate) in spleen, muscle, and liver, respectively (Table 1 and Supplementary Table 3). The higher number (%) of significant differentially used TSSs based on the Kolmogorov–Smirnov test (FDR *P*-value < 0.05) was observed between subspecies (71 (1.22%), 48 (1.33%), 600 (10.67%)) compared to within *taurus* subspecies (27 (0.46%), 11 (0.29%), 9 (0.28%)) in spleen, muscle, liver, respectively (Table 1 and Supplementary Table 3; Bootstrap-based *P*-value < 0.05). A list of genes with significant differential TSS usage with shifting score >0.1 across subspecies and *taurus* replicates is shown in Supplementary Data 4 and 5. Examples of genes with differentially used TSSs in different tissues across subspecies and within *taurus* subspecies is shown in (Figs. 2–4D; Supplementary Figs. 2–4D). Significantly enriched gene ontology (**GO**) analysis for genes having significant differentially used TSS across subspecies/biological replicates is shown in Supplementary Data 6 (*P*-value < 0.05). There were some overlap between two sets of enriched GO terms for a given tissue across subspecies/biological replicates.

Allele frequency changes could provide evidence for different selection pressure and differences in effective population size in two populations. In order to identify single nucleotide polymorphisms (SNPs) potentially differentially selected across subspecies among the TSS that were significantly different between the subspecies, we used the simple $F_{ST}$ based method (see Methods). The distribution of MAF for genome-wide SNPs with MAF > 0.005 in Brahman and Holstein animals available in 1000 bull genomes project were plotted (Supplementary Fig. 5A). As expected a more extreme allele frequency distribution was seen in Brahman than Holstein cattle due to a larger effective population size in the Brahmans[35]. The 472 SNP loci, among the TSS that were significantly different between the subspecies, had significant shifts in allele frequency between *Bos indicus* and *Bos taurus* subspecies based on pairwise $F_{ST}$ values and a Chi-square test combined with FDR less than 0.05 (Supplementary Data 7; Supplementary Fig. 5B). The distribution of the MAF of SNPs with significant shifts in allele frequency between subspecies (472 SNPs) and all SNPs, which were present in each of the populations (34,645,762 and 15,870,794 SNPs in *Bos indicus* and *Bos taurus*, respectively) were compared using a the hypergeometric test to evaluate the hypothesis that the MAF of SNPs with significant shifts in allele frequency among the significant differentially TSS regions will significantly differ from

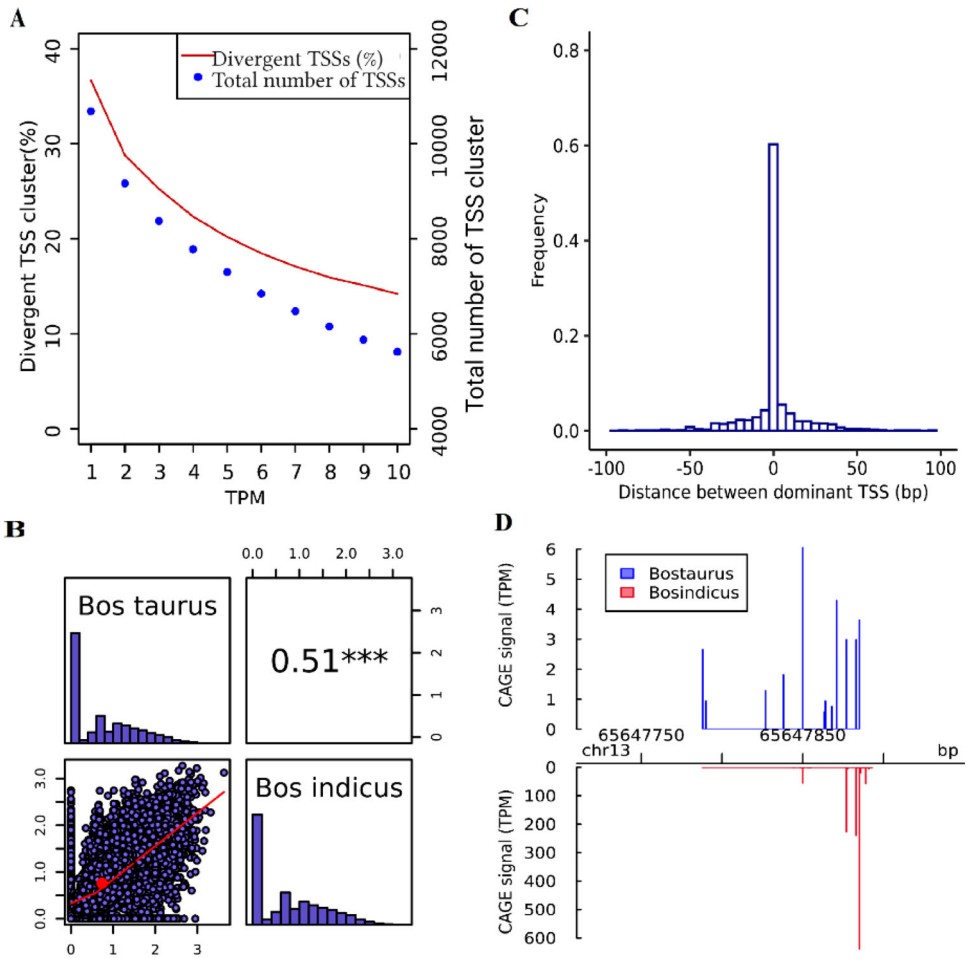

**Fig. 2 Overview of comparison between Bos taurus and Bos indicus subspecies for adult liver. A** Total number of TSS clusters and their percentage of divergent TSS at different expression levels (TPM), **B** Pearson correlation of TSS diversity measured by Shannon index (the distribution of Shannon index is shown on the diagonal and on the bottom of the diagonal the bivariate scatter plots with a fitted line are displayed. Also, on the top of the diagonal, the value of the correlation plus the significance level (*P*-value < 0.001) as stars is shown), **C** Histogram of distances between Bos taurus and Bos indicus dominant TSS in consensus TSS cluster with at least 10 TPM level expression, and **D** An example of differential TSS usage observed in gene MYL9 across two subspecies (*P*-value and FDR < 0.05; shifting score = 0.21). The annotated TSS based on the Ref-Seq gene using Apr.2018 (ARS-UCD1.2/bosTau9) is located at position 65,655,870 bp—(NM_001075234) myosin regulatory light polypeptide 9.

that of the entire genome (See Methods). In both subspecies, first two MAF bins were significantly more extreme to what was expected based on a random selection of SNPs from the whole genome (hypergeometric based test FDR *P*-value < 0.001, Supplementary Fig. 5C-D).

In line with evolutionary research from human and mouse[26,29] suggesting two types of evolutionary pathways lead to TSS turnover, we observed sliding of the TSSs along the genome and gradual shifting of usage from one TSS to an alternative TSS in the other subspecies. The differences observed in different tissues supported the tissue-specific effect of evolutionary changes in emerging new TSSs or degradation of a lowly expressed TSS over evolutionary time across the subspecies. Moreover, our findings suggested that evolutionary changes in TSSs might be tissue specific. Based on these results, one reason for the noticeable heterosis, which is observed in *Bos taurus × Bos indicus* crossbreds could be that there are more functional TSSs in crossbreds compared to the pure species. This mechanism for heterosis would have to be confirmed by demonstrating a dominance effect on fitness. Analysis of additional animals and potentially additional tissues relevant to divergent traits between the two subspecies would increase the resolution of our findings.

**Characterization of TSSs architecture in heat shock protein-related genes.** *Bos taurus* breeds are best suited to sub-tropical and temperate regions. They have thicker coats that allow them to endure cooler winters, and they do not have the notable 'hump' of their *Bos indicus* relatives. *Bos indicus* cattle in contrast have large ears and dewlap, which help to keep them cool and are well-suited to tropical environments. A previous study[36] revealed that the acute heat stress could increase the expression of heat shock proteins (HSP60, 70, and 90) and genes related to apoptosis (e.g., BAX, Bcl-2), suggesting that these genes have protective functions. With the hypothesis that heat shock proteins may be involved in this adaptation, 11 heat shock protein-related genes such as *HSP* family genes, including *HSP32*, *HSP60*, *HSP70*, *HSP90*, and *HSP105* (Supplementary Table 4) and an apoptotic-related gene (BAX, *BCL2*) were considered. The variation in dominant TSS position, and the distribution of TSS tags within TSS cluster in adult liver, spleen, and muscle tissues between *taurus* and *indicus* subspecies were evaluated for the above genes. The only differences observed between subspecies in terms of dominant TSS translocation was observed in genes *HSP5* and *HSP9* across adult liver tissues (shifting score 0.66 and 0.77, respectively; FDR *P*-value < 0.05; Supplementary Fig. 6A–E (I–III) and Supplementary Fig. 7A–D (I–III)).

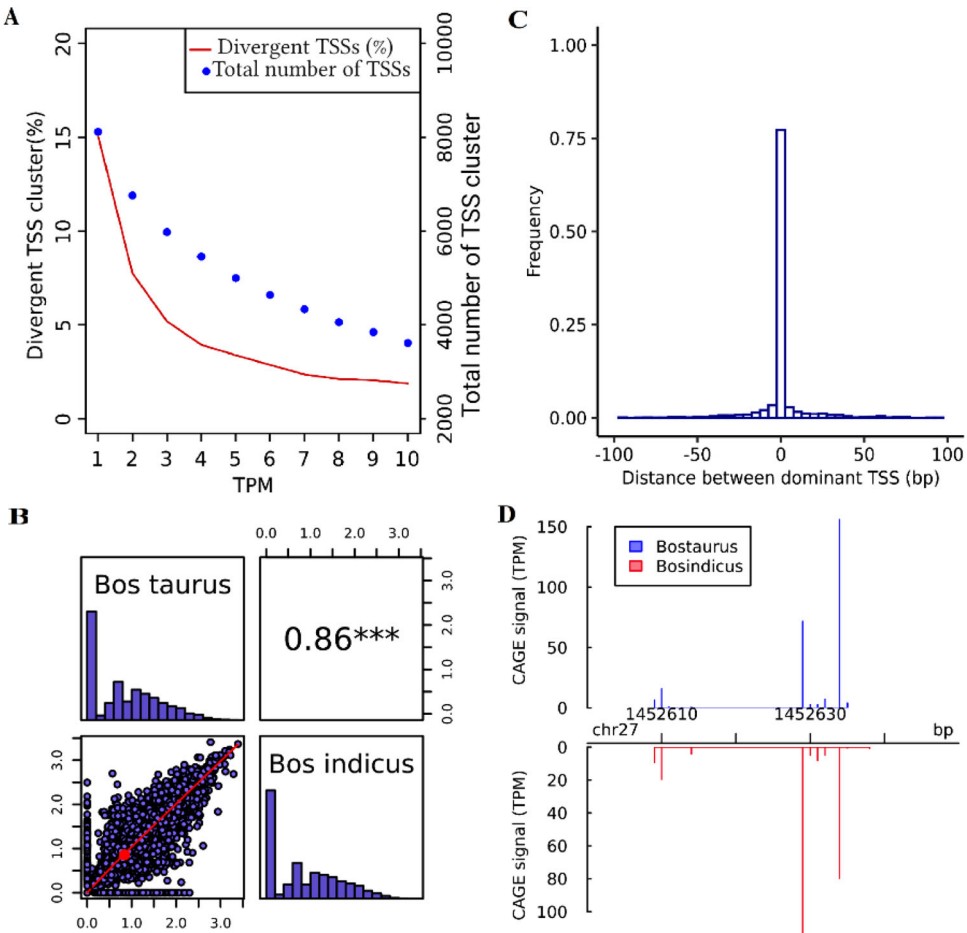

**Fig. 3 Overview of comparison between Bos taurus and Bos indicus subspecies for adult muscle. A** Total number of TSS clusters and their percentage of divergent TSS at different expression levels (TPM), **B** Pearson correlation of TSS diversity measured by Shannon index (the distribution of Shannon index is shown on the diagonal and on the bottom of the diagonal the bivariate scatter plots with a fitted line are displayed. Also, on the top of the diagonal, the value of the correlation plus the significance level (*P*-value < 0.001) as stars is shown), **C** Histogram of distances between Bos taurus and Bos indicus dominant TSS in consensus TSS cluster with at least 10 TPM level expression, and **D** An example of differential TSS usage observed in gene MYOM2 across two subspecies (*P*-value and FDR < 0.05; shifting score = 0.24). The annotated TSS based on the Ref-Seq gene using Apr.2018 (ARS-UCD1.2/bosTau9) is located at position 1,452,630 bp—(NM_001038140) myomesin-2.

All HSP genes we tested had higher TSS expression level in an *indicus* animal compared to *taurus* subspecies in spleen and muscle and the highest TSS expression was observed for gene *HSP90B1* in both subspecies (Fig. 5). TSS expression in genes related to heat stress across the *indicus* and *taurus* adult tissues (Fig. 5) and distribution of the TSS tags among the consensus TSS cluster (Supplementary Fig. 6A–E (I–III) and Supplementary Fig. 7A–D (I–III)) demonstrated association between TSS expression and shape in different genes. Consensus TSS cluster in the three highest expressed genes (*HSP90B1*, *HSP90AA1*, and *HMOX1*) were observed to have sharp, well defined TSS peaks, in which the distance between the 75 and 25 tag density percentiles within a TSS (i.e., interquartile range) was less than 4 bp (Supplementary Fig. 7A, B, D (I-III)).

**Variation in TSS usage between developmental stages**. The biological features of tissue in fetal and adult stages might be determined mainly at the level of gene expression. So differential and quantitative analysis of TSS expression and distribution patterns could be useful for the identification of developmental-stage-specific genes. The TSS were constructed separately for fetal and adult samples for *taurus* liver and *indicus* lung and liver, and then TSS clusters from fetal and adult samples were aggregated into a single set of consensus TSSs for each tissue (see the flowchart in Fig. 1). In total 16,549 and 19,149 consensus TSSs were detected in liver and lung *indicus* tissues and 15,345 in *taurus* liver, respectively (Table 2). About 67%, 60%, and 58% of those were located in the promoter of genes in *indicus* liver and lung and *taurus* liver, respectively (Table 2). In the pilot Encyclopaedia of DNA Elements study, it was shown that there are long transcripts that can bridge genes or even span several genes, often starting in the middle of a gene structure[37]. A small proportion of TSSs observed within exons (Supplementary Fig. 8) could be the result of recapping due to post-transcriptional modifications[37].

*Developmental-stage TSS changes in Bos taurus and indicus liver.* In both subspecies a lower TSS diversity was observed in the adult stage compared to the fetal (two Paired *t*-test with df = 11142 and 8966 in *indicus* and *taurus*, respectively, *p*-value < 2.2e-16). The average TSS diversity was about 0.97 (0.72) and 0.87 (0.78) in the fetal (adult) *indicus* and fetal (adult) *taurus* liver, respectively. Shannon indexes of TSS diversity have a tendency to be lower for the highly expressed gene than the relatively lowly expressed one and tend to rise with the number of TSSs in a cluster as well as the evenness of the relative uses of these TSSs[32]. A significantly

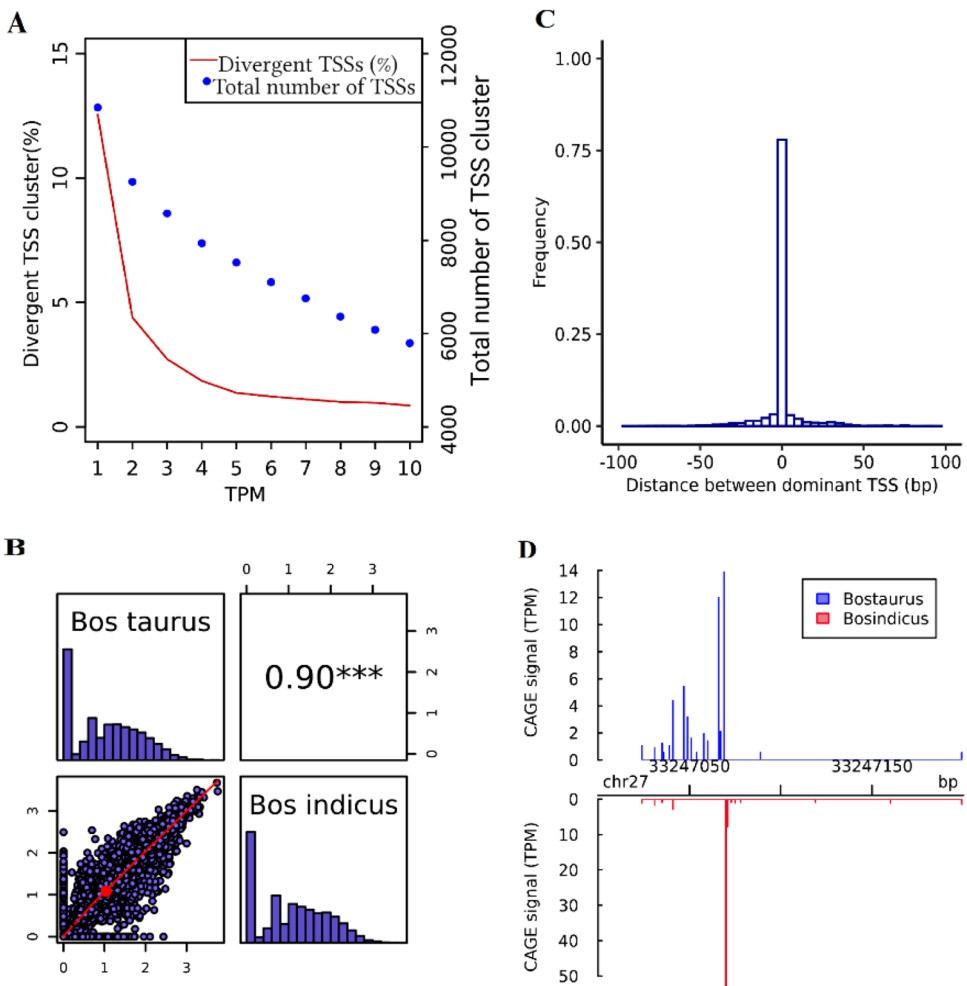

**Fig. 4 Overview of comparison between Bos taurus and Bos indicus sub-species for adult spleen. A** Total number of TSS clusters and their percentage of divergent TSS at different expression levels (TPM), **B** Pearson correlation of TSS diversity measured by Shannon index (the distribution of Shannon index is shown on the diagonal and on the bottom of the diagonal the bivariate scatter plots with a fitted line are displayed. Also, on the top of the diagonal, the value of the correlation plus the significance level (*P*-value < 0.001) as stars is shown), **C** Histogram of distances between Bos taurus and Bos indicus dominant TSS in consensus TSS cluster with at least 10 TPM level expression, and **D** An example of differential TSS usage observed in gene EIF4EBP1 across two subspecies (*P*-value and FDR < 0.05; shifting score = 0.86). The annotated TSS based on the Ref-Seq gene using Apr.2018 (ARS-UCD1.2/ bosTau9) is located at position 33,247,066 bp—(NM_001077893) eukaryotic translation initiation factor 4E-binding protein 1.

higher proportion of divergent TSSs across developmental stages was observed in *indicus* compared to *taurus* liver (14.7% vs. 6.4% of the total 5927 and 3938 of the consensus TSSs with at least 10 TPM) (Bootstrap-based *P*-value < 0.05, Fig. 6A(i-ii); Table 2). The total consensus TSS clusters in *taurus* and *indicus* liver tissues across fetal and adult stages are listed in Supplementary Data 8-9.

Similarly, when the position of dominant TSS for a given consensus TSS cluster with at least 10 TPM expression was compared between fetal and adult liver, the higher proportion of dominant TSS translocation across developmental stages was observed in *indicus* (40%) than *taurus* (25%) subspecies (Bootstrap-based *P*-value < 0.05; Fig. 6B(i-ii)). Consistent with this study, a previous study in human indicated that TSS switching events are common and can play a significant role in development[11]. The statistical significance of differential TSS usage was obtained by performing a two sample Kolmogorov–Smirnov test on cumulative sums of CAGE signal along the consensus TSS cluster. A list of genes with significant differential TSS usage with shifting score >0.1 across developmental stages is shown in Supplementary Data 10 (FDR *P*-value < 0.05 Kolmogorov–Smirnov test). One example of a gene with significant differential TSS usage between

fetal and adult stages in *taurus* is *SCD*, Stearoyl-CoA Desaturase. It is an enzyme in the chemical conversion of saturated fatty acids to unsaturated fatty acids. It is located on chromosome 26 in the region between 21,263,976 and 21,279,185 bp (ARS-UCD1.2; NCBI Ref-Seq Genes). Although the dominant TSS was found in the same position at the two developmental stages (21,263,953) in *taurus* liver, about 56% of transcription initiation in the fetal stage was happening downstream of the region used for transcription initiation in the adult tissue (FDR *P*-value < 0.05; Fig. 6C(i)). Another example of differential TSS usage observed in *indicus* liver is *SLC33A1*, acetyl-CoA transporter member 1 gene in *indicus* liver tissue which is located on chromosome 1 in the region between 111,829,274 and 111,850,037 bp (ARS-UCD1.2; NCBI Ref-Seq Genes). The dominant TSS was found in different positions, 111,829,274 and 111,829,278 bp in fetal and adult stage, respectively (FDR *P*-value < 0.05; Fig. 6C(ii)). The most enriched GO terms for genes with significant differential TSS usage expressed in *taurus* and *indicus* liver were *extracellular space, and structural constituent of ribosome.* (FDR *P*-value <0.05; Supplementary Data 11). The lower TSSs diversity correlation between fetal and adult stages was observed in *indicus* than *taurus* subspecies (Fig. 6D(i-ii)).

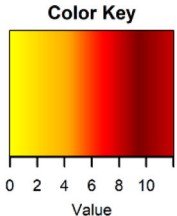

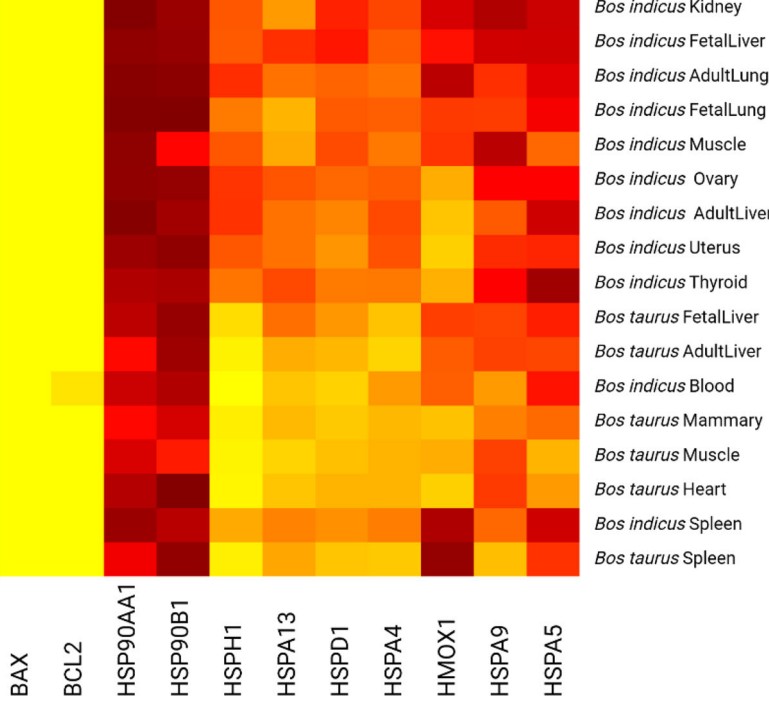

**Fig. 5 Consensus TSS cluster expression in genes related to heat stress across the Bos taurus tissues (heart, adult and fetal liver, mammary, muscle, and spleen) and the Bos indicus tissues (blood, spleen, thyroid, adult and fetal liver, adult and fetal lung, kidney, muscle, uterus, and ovary).** Log2 Tag per million (TPM) read was used as a measure of the expression level of RNAs in each tissue.

A crossover TSS event was identified if the dominant TSS switched between the two developmental stages for a specific gene (see Methods, scenario4). Only 388 genes having at least two consensus TSSs with at least 1 TPM expression in individual replicates were used. In total, 13 genes had crossover TSS switching events in *taurus* liver across developmental stages, which suggests variation in the dominant TSS over time (FDR $P$-value < 0.05 linear regression model; Supplementary Data 12). Since the alternative mRNA isoforms could be translated into functionally different products, a crossover switching event may suggest that one gene can play different roles at different time points in development. The most significant gene ontology term for genes with at least one crossover event were *RNA splicing* ($P$-value < 6.5E-2), *mRNA processing* ($P$-value < 8.3E-2).

*Developmental-stage TSS changes in* Bos indicus *lung.* Lung tissue plays an important role in the respiratory system of mammals after birth. Before birth, the lung is full of liquid[38–40] and does not participate in gas-exchange because of high pulmonary vascular resistance and immature respiratory function[41]. Therefore, it is necessary for the lung to be sufficiently developed at birth to perform the function of gas-exchange, which requires numerous physiological changes to occur[42]. We performed quantitative and expression analysis of CAGE-Seq data in fetal and adult lung tissues of a single *Bos indicus* cow-fetus pair. As expected, the proportion of divergent TSSs was reduced by increasing the TSS expression and mostly stabilized at the medium expression level

of 10 TPM (Fig. 6A(iii)). About 75% of TSSs clusters with at least 10 TPM expression had the same dominant TSS position across fetal and adult stages (Fig. 6B(iii)). The total consensus TSS clusters in *indicus* lung tissue in fetal and adult stages are listed in Supplementary Data 13.

Significant differential TSS usage occurred in 37 genes between the two stages (FDR $P$-value < 0.05 Kolmogorov–Smirnov test; Supplementary Data 11) and were enriched for the GO terms *extracellular region*, *focal adhesion*, *actin cytoskeleton*, etc (FDR $P$-value < 0.05; Supplementary Data 12). One example of a gene with differential TSS usage with aging in lung tissue is *WDR1*, WD repeat-containing protein 1, which is located on chromosome 6 in region between 105,290,992 and 105,332,584 (ARS-UCD1.2; NCBI Ref-Seq Genes). We found two dominant TSSs, which were 12 bp apart for this gene. The dominant TSS shifted with aging and about 26% of transcription initiation in the adult stage was happening outside of the region used for transcription initiation in the fetal lung tissue (FDR $P$-value < 0.05; Fig. 6C(iii)).

Interestingly in contrast to liver tissues, TSS diversity was higher in the adult compared to the fetus (Paired $t$-test, df = 11435, $p$-value < 2.2e-16). In total, the average TSS diversity was 0.84 (1.06) at the fetal (adult) stage in the lung. The correlation of TSSs diversity between fetal and adult stages was about 0.77 (Fig. 6D(iii)).

**TSSs discovery across adult tissues.** The total number of TSSs expressed (in promoter) in nine tissues from a single *Bos indicus*

**Table 2 Comparison of consensus TSS clusters in fetal and adult stages in *Bos indicus* liver, *Bos taurus* liver, and *Bos indicus* lung.**

| Tissue | Number of consensus clusters across developmental stages | | Number of consensus TSS clusters with at least 10 TPM expression level | | Number of divergent TSSs[a] | Number of significant TSSs with differential usage[a] with shifting score >0.1 | |
|---|---|---|---|---|---|---|---|
| | Total number | In promoter regions | Total number | In promoter regions | | Total number | Significant (P-value < 0.05) |
| *Bos indicus* | | | | | | | |
| Liver | 16,549 | 11,143 | 7,458 | 5,927 | 875 | 1,900 | 605 |
| Lung | 19,149 | 11,436 | 8,097 | 6,353 | 409 | 1,248 | 44 |
| *Bos taurus* | | | | | | | |
| Liver | 15,345 | 8,967 | 5,016 | 3,938 | 251 | 415 | 18 |

[a]With at least 10 TPM expression level in promoter region.

adult female and five tissues from three *Bos taurus* adult females were 48,473 (16,676) and 36,833 (12,150), respectively.

Generally, about 50–75% of CTSSs identified in different tissues were located in promoter regions (Supplementary Fig. 8). When the annotated TSSs based on the Ensemble release 102 or assembly release 106 were compared with those predicted in *Bos indicus* adult liver tissue, the majority of total genes expressed in adult *indicus* liver (75%) were located in close proximity (±35 bp) to annotated start coordinates. The remaining (25%) had TSSs at distances between ±35 and ±1016 bp of annotated start coordinates. Although there is some evidence that many novel core promoters especially for novel noncoding RNA are located in intergenic regions[37], for simplicity the final dataset was restricted to the promoter regions.

Interestingly, a noticeable proportion of the genes (21% and 24%) had divergent consensus TSS clusters, i.e., they had at least one TSS cluster not expressed in one or some tissues in *taurus* and *indicus* tissues, respectively. A higher proportion of divergent TSS clusters was observed between large discrepancies in coverage such as *Bos indicus* blood and liver (Supplementary Fig. 9A). As expected by increasing the expression level the proportion of divergent TSS clusters was reduced and mostly stabilized at the medium expression level of 10 TPM. For TSSs with at least 10 TPM expression, the highest and lowest proportion of divergent TSS were observed between liver-blood (17%) and ovary-uterus (1.3%) in *indicus*, and liver-muscle (17%) and heart-muscle (5%) in *taurus* subspecies, respectively (Supplementary Fig. 9A-B).

The highest (42%) and lowest (7%) proportion of differential TSS usage for TSSs with at least 10 TPM expression, was observed between lung–liver and lung–kidney in *indicus* tissues, respectively. While in *taurus* subspecies the highest (15%) and lowest (7%) proportion of differential TSS usage was observed between mammary–muscle and mammary–spleen, respectively. Our results highlight high potential for differential transcriptional regulation across tissues. This is in agreement with previous studies[4,43] indicating tissue-specific usage of TSS in mammals.

Tissues are distinguished by gene expression patterns, indicating distinct regulatory processes. Individual genes, or even sets of genes, in each tissue cannot adequately capture the diversity of structure and function that exist among different tissues[44], and multiple regulatory elements, including transcription factors and TSSs, that work together with other genetic and environmental factors must control the transcription of genes and production of proteins[45]. Alternative TSSs can result in higher or lower rates of protein synthesis[46,47]. When tissues were clustered based on their correlation between Shannon indexes of TSS diversity across adult tissues, they grouped mainly together into clusters reflecting their function (Supplementary Fig. 10(i-ii)).

Our results give some insight into how TSSs and frequent local insertions, deletions and duplications in the regions containing them can drive rapid evolution of species and subspecies. For example, duplication of TSSs can allow for neo-functionalization of genes, where an original gene takes on tissue-specific functions following the duplication event. This is similar to the neo-functionalization following whole genome duplication model proposed by a previous study[48], but on a gene scale rather than a genome scale. However, the TSSs duplication process will obviously be much more frequent, as demonstrated here, than whole genome duplication events.

## Conclusion

Knowledge of TSS expression and distribution would be a useful starting point to predict biological function of specific genes in different developmental stages or tissues. In the current study, we used CAGE-Seq data from *Bos taurus taurus* and *Bos taurus*

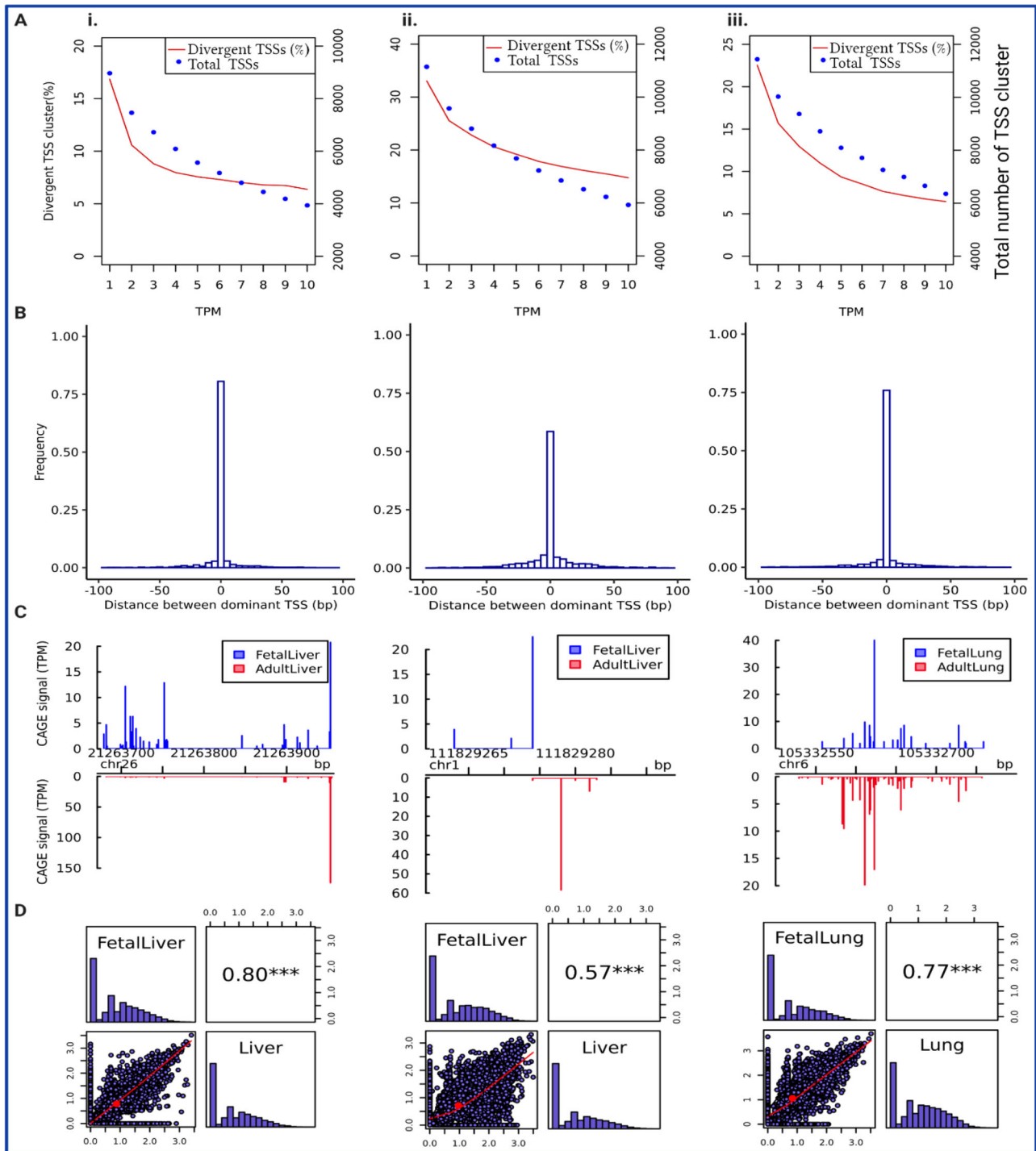

**Fig. 6 Overview of comparison between fetal and adult developmental stages for *Bos taurus* liver (i), *Bos indicus* liver (ii), and *Bos indicus* lung (iii).**
**A** Total number of TSS clusters and their percentage of divergent TSS clusters at different expression levels (TPM), **B** Histogram of distances between fetal and adult dominant TSS in consensus TSS clusters with at least 10 TPM level expression, and **C** An example of differential TSS usage observed in gene SCD in Bos taurus liver (shifting score 0.56), SLC33A1 in *Bos indicus* liver (shifting score 0.92) and WDR1 in *Bos indicus* lung tissue (shifting score 0.26) across two developmental stages (FDR *P*-value < 0.05). Shifting score is a measure of differential usage of TSSs within consensus cluster between two developmental stages, which indicates the degree of physical separation of TSSs used within given consensus TSS cluster. The annotated TSS based on the Ref-Seq gene using Apr.2018 (ARS-UCD1.2/bosTau9) for genes SCD, SLC33A1, WDR1 located at positions (21,263,976), (111,829,274), (105,332,584), respectively. **D** Pearson correlation of TSS diversity measured by Shannon index (the distribution of Shannon index is shown on the diagonal and on the bottom of the diagonal the bivariate scatter plots with a fitted line are displayed. Also, on the top of the diagonal, the value of the correlation plus the significance level (*P*-value < 0.001) as stars is shown).

*indicus*, diverged up to 0.5 million years ago, to assess changes in TSSs for these closely related cattle species. Also, we assessed developmental-stage changes in TSS usage and TSS shifting events in lung and liver tissues in cattle. Our results confirmed that TSSs evolve rapidly between species and even subspecies. The results of this study will accelerate future genomic research and will assist in narrowing down candidate genes with differential TSS usage. Our results also constitute an atlas of potential target sites (TSSs) for tissue-specific knockout or knockdown of gene expression with CRISPR/Cas9. A limitation with the current study is that only one biological replicate was included for the majority of tissues analysed in the study, so analysis of additional animals (and potentially additional tissues relevant to divergent traits between the two subspecies) would increase the resolution of the findings, particularly in the comparison of expressed TSS between the two subspecies.

## Methods

### CAGE library preparation and sequencing

Bos indicus. Thirteen samples from nine different tissues, including spleen, muscle, thyroid, ovary, kidney, uterus, lung, whole blood, and liver were collected from one pregnant Brahman cow and two tissues from the cows fetus, including lung and liver were collected in accordance with relevant guidelines and regulations approved by the Queensland Department of Agriculture and Fisheries Animal Ethics Committee.

Whole blood samples (500 µl) were collected in RNAprotect Animal Blood tubes (QIAGEN). All other samples were harvested after the animal was slaughtered and immediately snap-frozen in liquid nitrogen and stored at −80 °C until processing.

Frozen tissues were grounded in liquid nitrogen with mortar and pestle. Total RNA was extracted from ~50 mg ground powder using the mirVana miRNA Isolation Kit (Ambion, Carlsbad, CA, USA) according to the manufacturer's instruction. Due to the yellow colour of extracted RNA from the thyroid and spleen, 60 µl of each RNA solution was aliquoted and precipitated using Lithium Chloride (Sigma–Aldrich). Both the unprecipitated and precipitated thyroid and spleen RNA was used for CAGE library preparation.

The extracted RNA was quantified using QubitTM 4.0 Fluorometer and the Nanodrop ND-1000 spectrophotometer (v.3.5.2, Thermo Fisher Scientific). Depending on the RNA concentration, RNA integrity was determined by Agilent RNA 6000 Nano or Pico kits on the Agilent Bioanalyser 2100 (Agilent technologies, Belgium). RNA integrity numbers (RIN) ranged from 8.1–8.5 for all RNA samples.

Bos taurus. Four lactating cows, two of which were pregnant (16 weeks gestation) were selected from the Agriculture Victoria Research dairy herd at Ellinbank. Following euthanasia mammary gland was collected from all four cows, heart, liver, spleen, and semimembranosus muscle was collected from the two pregnant cows at the Ellinbank research facility with approval from the DEDJTR Animal Ethics Committee (2014-23). Cows were individually restrained in a crush and given an intravenous injection of 10% zylazil adequate to cause moderate sedation. Each cow was then immediately released from the crush and upon the cow laying down a veterinarian euthanized the animal by lethal injection, using Pentabarb (sodium pentobarbitone 200 mg/ml) administered intravenously at dose rates greater than 100 mg/kg until the cow was deceased. Once pronounced dead all tissue types were dissected from the animal. Connective tissue was removed and the samples dissected into 1 cm cubes, sealed in a 5 ml tube and flash frozen in liquid nitrogen and subsequently stored at −80 °C. Tissue samples were ground using a Genogrinder (SPEX SamplePrep), keeping samples frozen by using liquid nitrogen cooled tubes and tube racks. Up to 50 mg of ground tissue was used for total RNA extraction using the *mir*Vana miRNA Isolation kit (Ambion) according to manufacturer's instructions. The extracted RNA was quantified using QubitTM 3.0 Fluorometer (Invitrogen) using the Qubit RNA BR Assay Kit (Invitrogen) and the Nanodrop 1000 spectrophotometer (Thermo Fisher Scientific) according to manufacturer's instruction. RNA concentration and integrity were determined on the Agilent Tapestation (Agilent technologies) according to manufacturer's instructions. RNA integrity numbers (RIN) ranged from 6.1 to 7.7 for all RNA samples.

CAGE libraries were constructed from total RNA samples as described previously[19]. Briefly, 27-nt long tags corresponding to initial bases at the 5' end of capped RNAs were prepared with barcode linker sequences and the necessary adapters to allow sequencing on Illumina single-end flow-cells. Barcoded CAGE libraries were pooled in sets of 8 prior to PCR amplification to minimize amplification bias. Libraries were prepared and sequenced at the Western General Hospital, Edinburgh. The sequencing was done using an Illumina HiSeq 2500 platform (50nt singl-end).

### Read processing and alignment

Sequence read quality was assessed using the FastQC[49], including calculation of GC content, and identification of over-represented sequences. The EcoP15I fingerprint was trimmed by cutting the first 9 bases (*CROP:9*) and Illumina adaptor trimmed by cutting the end 14 bases (*HEADCROP:36*) using Trimmomatic[50] (version 0.35). Trimmed reads were aligned to *Bos taurus* reference genome (GenBank: ARS-UCD1.2) with Burrows-Wheeler Aligner (**BWA**[51], version 0.7.13) using the BWA-MEM algorithms. The aligner was run using default parameters, the only exceptions were t = 10, and k = 10. Also, to alleviate the presence of universal G at the head of the read, which may be present in some of the reads, parameters L (clipping penalty) and B (mismatch penalty) were assigned as 4 and 5, respectively.

### Simulation

The selection of an appropriate alignment tool for CAGE-Seq data can be a difficult due to their short read length. Therefore, we simulated simulated single-end sequence datasets with read lengths of 27 bp similar to the length of the trimmed CAGE-Seq data and compared alignment quality of BWA and Bowtie2[52] (version 2/2.3.4.3). Simulated datasets were generated from chr1 of the *Bos taurus* genome (GenBank: ARS-UCD1.2) using dwgsim[53]. The default per base sequencing error rate of 0.02 was considered. Three datasets, each comprised of 20 samples, were generated with average sequencing depth of 10–25x (high), 5–10x (medium), and 1–5x (low). The sequencing coverage of each sample for each datasets was chosen based on random distribution within the coverage bounds. All simulated reads were mapped to chr1 of the *Bos taurus* genome assembly.

The parameters used for running BWA was the same as the parameters used in real data. For Bowtie2 the default parameters were used. Two standard performance measures, precision, and recall were used to evaluate the aligners. Recall (sensitivity) indicates the number of correctly aligned reads over the total number of reads that should have been aligned, and precision shows the number of correctly aligned reads over the total number of aligned reads. The measures were calculated using the dwgsim_eval program dwgsim[53]. To assess the overall performance of the two aligners, the area under the precision-recall curve (**PR-AUC**) was computed. PR-AUC ranged between 0 and 1 with larger area indicating better performance. Overall scoring of the mappers based on our evaluation criteria was slightly higher for BWA compared to Bowtie2 (0.41 ± 0.0 vs. 0.32 ± 0.0008), indicating the higher accuracy using BWA with respect to sequencing parameters used.

### Quality controls and preliminary analyses

Only primary alignments with a quality of greater than 20 (<99% chance of true) were considered for TSS calling. Furthermore, the CAGE tag starting sites (CTSSs), a nucleotide position on the genome from which an alignment of CAGE tag starts, supported by 3 or more CAGE read 5'-ends in a single sample were selected. The total number of reads before and after quality control and selected numbers of supporting CAGE tags in each tissue is shown in Supplementary Table 1. The flowchart illustrating the entire flow of the analysis is shown in Fig. 1. Different scenarios were investigated based on the scope of the analysis and the way biological replicates/tissues were taken into account:

*Scenario1*

## Evolutionary TSSs discovery

The pairwise comparisons between *Bos taurus* and *Bos indicus* subspecies for spleen, muscle, and adult liver were carried out by following the workflow in Fig. 1. For this scenario, *Bos taurus* biological replicates for each tissue were merged together resulting in a single sample that contained a union of CTSSs present in the replicates and raw tag counts for those CTSSs. Then, TSSs were called in each subspecies tissue and then TSSs from *Bos taurus* and *Bos indicus* samples were aggregated into a single set of consensus TSS clusters for each tissue. Different TSS properties, including the proportion of divergent TSSs (i.e., TSSs observed only in one subspecies), distance distribution between dominant TSSs across subspecies, and shifting of usage from one TSS to an alternative TSS in the other subspecies were compared (see Differential TSS usage and shifting score calculation).

To investigate the changes in Minor Allele Frequency (MAF) in *Bos taurus* population vs. *Bos indicus* population in significant differential TSS regions, all variants within significant differential TSS cluster regions across subspecies was extracted (2,926 loci) from 200 Brahman (*Bos indicus*) and 1053 Holstein (*Bos taurus*) animals available from 1000 bull genomes project (Run8). To identify SNP loci that had been under selection, pairwise $F_{ST}$-values comparing *Bos taurus* and *Bos indicus* was calculated for SNPs with MAF greater than 0.005 using Weir and Cockerham's method[54] implemented in vcftools (version 0.1.13)[55]. Then, a Chi-square test was used to identify SNPs with significant $F_{ST}$ at *P*-value < 0.05, using the test statistic $X^2 = 2 NF_{ST}$, where $2N$ = the sum of genotyped gametes in the two populations[56]. Finally, corresponding overall estimates of the false discovery rate (FDR) were calculated for each significant SNP as l*P/d, where l = number of SNP loci tested (667 SNPs), P = the significance level of the individual chi-square tests, and d = the number of SNP loci identified with a significant $F_{ST}$ (475 SNPs). To test for differences between the MAF of significant SNPs identified using $F_{ST}$ method (FDR < 0.05) within significant differential TSS cluster compared to within the whole genome, a hypergeometric distribution test was used (*P*-value < 0.01). The difference in MAF within

each of 50 bins, where each bin contained a 1% range of MAF values was tested using hypergeometric probability density function:

$$p(x, N, n, m) = \frac{\binom{m}{x}\binom{N-m}{n-x}}{\binom{N}{n}} \quad (1)$$

where $p(x, N, n, m)$ is the probability of observing $x$ SNPs within a given MAF bin in a sample of 472 SNPs ($n$) drawn from a *Bos indicus* (*Bos taurus*) genome-wide of 34,645,762 (15,870,794) SNPs ($N$) containing $m$ SNPs within a given MAF bin. The values of $x$ are defined based on the number of significant SNPs identified using $F_{ST}$ method (FDR < 0.05) in a given MAF bin for each subspecies. The value of $m$ are obtained based on the number of SNPs observed in a given MAF bins for each subspecies. This was done for all 50 MAF bins for both subspecies and P-values were adjusted for multiple testing (FDR < 0.001).

*Scenario2*

## Subspecies TSSs discovery (within *Bos taurus*)

Due to the lack of biological replicates, the experimental design does not allow support of the conclusions from the evolutionary comparative analyses. Consequently, the reported differences obtained from the scenario1 cannot be reliably assigned to the factor of interest (the subspecies). To support our conclusion from evolutionary TSS discovery (scenario1) and get insight about expected variation within subspecies, pairwise comparisons were implemented between the *taurus* biological replicates for spleen, muscle and liver. TSSs were called in each biological replicate in three tissues and then TSSs from *two* biological replicates were aggregated into a single set of consensus TSS clusters for each tissues (Fig. 1). Similar to first scenario, the proportion of divergent TSSs (i.e., TSSs observed only in one subspecies), distance distribution between dominant TSSs across biological replicates, and shifting of usage from one TSS to an alternative TSS in the other replicate were compared (see Shifting score measurement paragraph below). This could estimate the expected difference observed between samples from the same subspecies, and potentially help to compare samples from the *Bos indicus* animal.

*Scenario3*

## Developmental-stage TSSs discovery

The pairwise comparisons between fetal and adult stages for *Bos taurus* liver, *Bos indicus* lung and liver were done by following the workflow in Fig. 1. Similar to first scenario, the *Bos taurus* biological replicates were merged together before TSS calling. Then TSSs called in fetal and adult stage were aggregated into a single set of consensus TSS clusters separately for each tissue and different properties of TSSs between fetal and adult stages were compared.

*Scenario4*

## Discovery of crossover TSS switching events across fetal and adult *Bos taurus* liver

To investigate the crossover switching event, TSSs called in individual fetal and adult *taurus* replicates (four samples for each tissue) were aggregated into a single set of consensus TSS clusters. Only genes having at least two consensus TSSs with at least 1 TPM expressed in both replicates were used. The null hypothesis was that there was no switching for the two TSS clusters. The test of this hypothesis was performed using *lm()* function in R (version 4.0.2) as follows and candidate switching events identified at this preliminary stage if the p-value for interaction term was less than 0.01.

$$Y = \alpha + \beta1*TSS + \beta2 \times stage + \beta3 \times TSS*stage \quad (2)$$

where Y represents the TSS expression, TSS represents the effect of the consensus TSSs (presumed here to be 1 and 2), stage is the effect of developmental stage (fetal and adult), and the interaction term between TSS and stage.

A crossover TSS switching event was detected if one TSS cluster was used more frequently at one developmental stage compared to the other, and that the dominant TSS cluster switches between fetal and adult stages. Corresponding overall estimates of the false discovery rate (FDR) were calculated for all significant genes based on the number of gene tested (388 genes), the significance level of the individual linear regression model for interaction term, the number of genes with crossover TSS switching event.

*Scenario5*

## Tissue-specific TSSs discovery

For this scenario, biological replicates were merged before TSS calling for *Bos taurus* subspecies. The TSS were called for each tissue and then to be able to compare samples at the level of clusters of TSSs between tissues, TSS clusters from all samples were aggregated into a single set of consensus TSS clusters (Fig. 1) separately for each subspecies. Then, for each supspecies (1) the distance distribution between dominant TSSs, (2) the

proportion of TSS clusters with differential TSS usage, and (3) significant 'shifting' TSS clusters for pairs of tissues was estimated.

**Tag count normalization**. To enable comparison between multiple samples raw tag counts were normalized. It has been shown that many CAGE datasets follow a power-law distribution[31]. Therefore, plotting the number of CAGE tags against the number of CTSSs that are supported by that number of tags can help in choosing the best range of values for a distribution that can be approximated by a power-law. The slope of the suggested reference distribution (alpha), calculated by the median of slopes fitted to individual samples, are shown in Supplementary Fig. 11A–D (1-3). The T parameter was selected to be $10^6$, so normalized values were expressed as tags per million (TPM). The normalization process was run in CAGEr[34] (Version: 1.32.0).

**TSS calling and clustering**. TSSs clustering was performed using a simple distance-based clustering using 20 bp as a maximal allowed distance between two neighbouring TSSs in CAGEr. Prior to clustering low-fidelity TSSs, those supported by less than 1 normalized tag counts were filtered out. To be able to compare genome-wide TSSs across samples, TSS from specified samples were aggregated into a single set of non-overlapping consensus TSS clusters (different samples were choose based on the 5 scenarios mentioned above). Two clusters were aggregated together if their boundaries (i.e., 0.1 and 0.9 positions of quantiles) were less than 100 bp apart.

**TSS annotation**. The function *annotateCTSS()* implemented in R and ENSEMBL genome reference *Bos taurus* ARS-UCD1.2 annotation (release 102) were used for the annotation of the CAGE profiles. Only consensus TSSs located in the promoter regions were used for further analysis. To demonstrate the provided functionality of various outputs produced in the current study, the predicted TSSs in the more robust tissue (*Bos indicus* adult liver) was searched against Ensemble annotation (release 102) and assembly ARS-UCD1.2 (release 106).

**Differential TSS usage and shifting score measurement**. Consensus TSS clusters within the same promoter region could be used differently in different samples, while having the same overall transcription level. So, the differential usage of TSSs (*promoter shifting*) can reveal changes in the regulation of transcription between two samples, which cannot be detected by expression profiling. Shifting score as a measure of differential usage of TSSs within consensus TSS cluster between two samples was calculated for all consensus clusters between two specified (groups of) CAGE datasets using *scoreShift ()* function in CAGEr. Shifting score indicates the degree of physical separation of TSSs within a given consensus TSS cluster and was calculated as score = max(F1 − F2)/max(F1), where F1 is a cumulative sum of CAGE signal along the consensus TSS cluster in the sample with the lowest total signal in that consensus cluster, and F2 in the other sample. Shifting score was calculated for both forward (5'–>3') and reverse (3–>5') direction and the bigger value was selected as the final shifting score. The statistical significance (P-value and FDR) of differential TSS usage was calculated based on Kolmogorov–Smirnov test using *useTpmKS = TRUE* function in CAGEr.

**Shannon index of TSS diversity**. To quantify the TSS diversity of each TSS cluster, the Shannon index was measured as $-\sum_{i=1}^{S} P_i ln P_i$ for each consensus TSS cluster, where S is the number of individual TSSs among the given consensus TSS cluster and $P_i$ is the proportion of CAGE signal along that consensus TSS cluster corresponding to the *i*th TSS.

**Enrichment gene ontology (GO) analysis**. For genes having at least one significant differentially used TSS or crossover switching event between two samples GO analysis were carried out by using the DAVID[57] web server (version 8) and for each comparison the reference was defined as the set of all expressed genes for that analysis.

**Bootstrapping-based test**. A bootstrapping analysis was conducted to compare the chance that the differences in the observed proportions of TSS positions was due to random chance. The bootstrap was run by randomly selecting a TSS position from all observed positions in that region, where the probability of selecting each position was equal to the proportion in the observed sample. This was repeated 10,000 times. The distribution of TSS in each of the positions was then compared to the other samples. If the observation in the other samples was outside the 95th percentile of the randomly selected distribution, the observed differences between samples was determined as not due to random chance.

**Statistics and reproducibility**. In this study we exploit a much closer evolutionary split between *Bos taurus taurus* and *Bos taurus indicus* using CAGE datasets to assess changes in TSSs for closely related (cattle) species, with the aim of gaining further insights into the evolution of TSSs. One Brahman adult and fetus (*Bos indicus* subspecies), and four Holstein lactating cows (*Bos taurus* subspecies), including two pregnant (16 weeks gestation) and their fetus were used. CAGE-Seq

was performed on 11 tissues at adult stages, including liver, lung, kidney, thyroid, spleen, muscle, uterus, ovary, blood in *indicus* and liver, spleen, muscle, mammary, heart in *taurus* subspecies, and two tissues in fetal stage, including liver and lung in *indicus* and liver in *taurus* subspecies. Technical reproducibility was undertaken by splitting the data into lower depth and determining if the same calls were made. The comparative analysis was done using CAGEr between two subspecies in the three tissues (adult spleen, muscle, and liver), along with its internal control between biological replicates of the same subspecies (*taurus*). Also, the comparative analysis was done between developmental stages in two subspecies. The bootstrapping analysis was conducted for calculating significance level.

**Reporting summary**. Further information on research design is available in the Nature Research Reporting Summary linked to this article.

## Data availability

*Bos taurus* and *Bos indicus* raw sequence data are publicly available via European Nucleotide Archive (ENA) under study ID PRJEB43513 and PRJEB44817, respectively. Sample metadata for *Bos taurus* is available in the BioSamples database under accessions SAMEA8326848, SAMEA8326850, SAMEA4447839, SAMEA4447825, SAMEA4447799, and SAMEA4447832. Sample metadata for *Bos indicus* is available in the BioSamples database under accessions SAMEA8976600, SAMEA8976601, SAMEA8976602, SAMEA8976603, SAMEA8976604, SAMEA8976605, SAMEA8976606, SAMEA8976607, SAMEA8976608, SAMEA8976609, SAMEA8976610. All other relevant data are available in this article and its Supplementary Information files. See Supplementary Table s1–4, Supplementary Data 1–13, and Supplementary Figs. 1–11 for extended biological findings.

## Code availability

Preprocessing of CAGE sequencing data, identification and normalization of transcription start sites, and downstream analysis of transcription start sites clusters (promoters) were done using CAGEr, an R package for CAGE data analysis (https://bioconductor.org/packages/release/bioc/vignettes/CAGEr/inst/doc/CAGEexp.R). All statistical analyses were performed using R version 4.0.2.

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

## Acknowledgements

We acknowledge financial contributions from Meat and Livestock Australia (Project P. PSH.0868—"Characterisation of the Brahman Genome") for the generation of the *Bos taurus indicus* CAGE-seq data. We would also like to acknowledge financial contributions from DairyBio (a joint venture project between Agriculture Victoria and Dairy Australia) and Research Initiative Fund of the Faculty of Veterinary-& Agriculture Sciences of The University of Melbourne for the generation of the *Bos taurus taurus* CAGE-seq data. We are thankful to Dr. Brian Burns for helping source the *Bos taurus indicus* tissues, and Dr. Bronwyn Venus for collecting *Bos taurus indicus* samples. Thank you to Elise Kho for extracting some the Bos taurus indicus RNA. We are also thankful to Rodger and Lorena Jefferis of Elrose Brahman Stud who donated Neomi the Brahman cow used in this study to science.

## Author contributions

M.F., E.R., B.H., A.C. and R.X. conceived and designed the experiments. L.N. and B.M. prepared the RNA. M.F. analysed the data. J.G. was responsible for the collection of *Bos taurus* samples. S.M. led the project that investigated the Brahman genome which led to this work. M.F. wrote the paper and all authors revised it critically for important intellectual content. All authors read and approved the submitted manuscript.

## Competing interests

The authors declare no competing interests.
