## [Peer Review File · Communications Biology]

Reviewers' comments:

Reviewer #1 (Remarks to the Author):

This manuscript presents an analysis of CAGE sequencing data that were obtained across a large set of tissues in two cattle subspecies: *Bos taurus* and *Bos indicus*.

The analysis starts with typical steps like read mapping and peak calling, which generates a set of predicted TSSs. After several filtering steps, the number of TSSs per gene is computed and compared between subspecies. Shapes of the TSSs are also considered. Expression levels are assigned to TSSs with a focus on Heat Shock Protein genes (HSPs) and a hierarchical clustering of tissues is realized based on these values. TSS numbers per gene are again compared, this time between adult and fetal tissues, and used to correlate tissues.

Overall, many comparisons and observations are made using this data but most of them lack a proper assessment of statistical significance. This is mostly due to an incompatibility between the experimental design and the analysis approach. This is a strong limitation to draw reliable conclusions from the results. I believe that the analysis could greatly be improved, in particular by changing the way replicates are taken into account. At several occurrences the methods should be clarified, better documented and cleared of potential biases. The results section is a bit repetitive and would benefit from more biological relevance and novelty. The CAGE sequencing data itself is of interest to the community though.

My review is divided in three parts: (A) Major general issues, (B) Detailed comments, and (C) Minor issues.

A. Major general issues

1) Lack of significance

Due to the lack of replicates, the experimental design does not allow to support the conclusions from most of the comparative analyses. For instance, since only one adult cow and one fetus are available in the *B. indicus* group, the comparison with the *B. taurus* group is strongly limited, in particular with such a little set of common tissues. Consequently, the reported differences cannot be reliably assigned to the factor of interest (the subspecies). Same goes for the adult vs. fetus comparison, with only one and two fetuses from the *indicus* and *taurus* subspecies respectively.

I would suggest the authors to strengthen their experimental design, adjust their claims, or change the scope of the analysis. Instead of using the replicates to discard an unreported proportion of the called TSSs, one way to proceed could be for instance to estimate for each tissue the distance distribution between proximal TSSs from the available duplicates in the *B. taurus* group. This could give an estimation of the expected difference to be observed between samples from the same subspecies (cf. Fig.1), and potentially help to compare samples from the *indicus* animal. The liver tissue is actually the only one with biological replicates in all the groups (one adult and one fetus from *indicus*, one fetus and two adults from *taurus*).

An alternative way to design the analysis would be to focus on the multiplicity of available tissues to better characterize the diversity of TSS usage and explore regulatory aspects through a rigorous analysis of a tissue-specificity criteria for instance. This would better fit the available data than a pairwise-oriented comparative approach between groups with insufficient numbers of replicates.

2) Issue of the "TSS number per gene" metric.

A large part of the results are based on comparing the number of TSS per gene between groups of

samples. This number depends on the total number of predicted TSS, which is explicitly adjusted by the authors for each tissue during the peak calling step (with the threshold parameter t_i).

First, it does not seem relevant to me to normalize read coverage by adjusting the number of predictions, in particular when it is done by raising a threshold detection in abundant datasets. This might lead to a drastic and unnecessary loss of data by artificially reducing the number of detected TSSs in large libraries. There is nothing wrong in obtaining more TSSs when sequencing deeper, it is a natural way to detect transcripts with low expression levels. So as long as noise saturation is not reached and called peaks indicate genuine TSSs (which is not not addressed here), there is no reason to remove valid predictions using the worse sample as a reference. Transcriptome studies based on RNA-seq do not reduce the number of predicted genes/transcripts to compare samples but use instead proper coverage normalization methods. In addition, removing TSSs that do not appear at the exact same position in several replicates as indicated in the Sup. Methods (p.4, l.85) is certainly excessive. The CAGEfightR manual explicitly recommends to use the "support" threshold for cases with "a very large number of and/or very noisy samples (For example due to poor RNA quality)", which does not seem to be the case here.

Second, the number of TSS per gene depends on several confounding factors that are not controlled, including the amount of sequencing data (number of tags) and the expression level of the genes. It might be for instance that an observed difference in number of TSSs is uniquely due to a global increase of gene expression level, which allows for a more comprehensive profiling of the transcriptome complexity while no active biological regulation could actually be involved at the level of TSS usage. These confounding factors need to be properly investigated and addressed before considering the number of TSS per gene as a reliable metric.

3) Issue of the GO analysis

A Gene Ontology (GO) term enrichment analysis needs a carefully defined gene set to be provided as a reference background. When no relevant gene set of reference is available, it is possible to provide the computational tool with the entire set of known genes for that species. This is what has been done in that study (p.4, l.91 of the Sup Methods: "Bos taurus genome was considered as background in all analyses"). In this case however, as in many transcriptome studies, the reference should be defined as the set of the expressed genes, not the set of all known genes. For instance, when looking for GO term enrichment in a subset of differentially expressed genes, the reference background needs to be the set of the expressed genes, including the differentially and the non differentially expressed ones, but excluding non expressed genes that were not tested for differential expression. Similarly, in this study, when performing a GO term enrichment analysis on a subset of genes, for instance expressed genes "with a single TSS" (p.8 l.213), the background must be limited to the set of all expressed genes, including the ones with a single TSS and the one with several TSSs, but not the ones with no expression and consequently no predicted TSS. Otherwise it is not possible to define if the GO term enrichment is due to the factor of interest (the specific subset) or to the simple fact that the subset of genes belongs to the larger set of expressed genes, and any random subset could lead to the same enrichment. All GO analyses should be fixed accordingly.

4) Gene expression analysis

Gene expression analyses are incorrect or insufficiently documented. For instance p.7, l.184 ("t-Test") or p.10, l.252. ("We performed quantitative and expression analysis of CAGE-Seq data", with no further detail). It should be noted that the CAGEfightR package used in this study generates output tables that are compatible with proper gene expression analysis tools like edgeR or DESeq2.

5) Novelty and biological significance of the results

Differences are reported between samples and some variation is observed across tissues, but all this

seems quite expected to me and do not bring new conclusions considering existing reports on human/mouse. The data is original and has potential, but no strong biological results is clearly drawn from the study.

The comparison with a reference annotation like REFSEQ or ENSEMBL is curiously missing. The distribution of the predicted TSSs in the diverse types of genomic regions should be provided, including the proportion of TSSs in promoters, near an annotated TSS, in 5'UTR, etc. Also, the distance distribution of the predicted TSSs with respect to the annotated ones would greatly help to show the value of the data.

CAGEfightR was used to call the TSSs but apparently not to detect enhancer RNAs (using the bidirectional clustering function of the package). Is there a reason for that? Considering the broad set of tissues in this study, identifying enhancers and, if possible, the tissue-specific or ubiquitous ones, might provide valuable results and biological insights.

7) Several steps of the analysis are not clear or insufficiently documented. Several statistical tests are performed without providing enough details about the size of the groups, the exact numbers or proportions that are compared, etc. The writing should be improved to better explain the methods (see detailed comments below).

8) Last but not least: data availability. Sequencing data must be available to allow publication. Sequences have to be submitted to a dedicated public database (SRA, ENA) and the accession codes provided in the manuscript. The data availability statement "For data availability please contact the authors." in the Reporting Summary is not acceptable.

B. Detailed comments

About the "statistical analysis of the reproducibility" (page 3, bottom).

- From Table 1 and S1, it seems that there is a positive correlation between the number of CAGE tags and the number of called TSSs. This is an important point that should be addressed and commented in the text.

- The impact of the number of replicate, however, was not obvious to me at all. Since no results from TSS calling are presented on individual replicates, I first thought that replicates were simply merged together before TSS calling, which is an acceptable strategy once replicates are shown to be consistent. The only clue I could find about the process was one brief mention in the supplementary note: "TSSs appearing in only a single replicate were discarded" (p.4, l.85). Does it mean that only TSSs found exactly at the same position in different replicates were kept? This would imply that two adjacent TSSs for instance would be both discarded and none of them would be considered. Is that correct? This seems like a very stringent and restrictive filter to me: how many TSSs have been discarded like this?

- If replicates have been processed separately please clarify in the text and provide the raw results in Table 1 (the number of called TSSs for instance are missing for replicates 2 and 3) and in Supplementary Table 1 as well (number of TSSs in each biological replicate and subsamples). Please also clarify the merging process of the results from the replicates and the subsequent outcomes, in particular when 3 replicates are available.

- "for samples with high coverage [...], to avoid missing a noticeable number of true TSSs, the t_i was set to 25 tag counts" (p.4 l.87): this sounds strange to me because missing true positive is more likely in samples with low coverage, not high coverage.

- Assessing the reproducibility based on the sole number of predicted TSSs is not satisfying. The proportion of identical/proximal predictions would be more informative. Consider a Venn diagram for instance. If only the number of predictions is considered and not their nature, the statement of 80% reproducibility has to be removed (p.4, l.94). Showing a strong correlation/consistency between replicates would support the general reliability of the data. Idem for subsampling results: how close are TSSs predicted from subsamples?

The part on classifying TSSs into shape categories (p.5 l.119) is not clear and needs to be rewritten.

- A detailed presentation of an existing classification is first made. Then it is difficult to understand why the authors do not follow that classification. The sentence "[...] but in the current study to better visualize the probability distribution of the cluster of tags within a TSS we classified TSS into four shape categories based on the genomic interval covered by the cluster of tags and the probability of the tags distribution within a single TSS" is not consistent with the proposed classification. "In the single dominant peak class (SP), the tags were concentrated to no more than four consecutive start positions." Is that an observation using the previous definition of the SP class or is it a new definition of the class? How is that observed exactly or applied in practice? If it is a new criteria to define the SP class, why is the former criteria of the IQR size used in the next sentence ("The clusters spanning a broader region (IQR>4 bp)")?

- Figure 3: the proportion of the TSS in each category is then compared between subspecies within groups of TSSs. No statistical significance is proposed to support the potential relevance of these observations. According to the figure, the TSS distribution within categories seems highly variable, so even comparing random subsamples might lead to very different proportions. Biological relevance is also missing from these observations. For instance I do not see what conclusion could be drawn from an observation like "TSSs in indicus tended to be wider, spread over a larger genomic region, in genes with divergent TSSs in which indicus and taurus sub-species had single and multiple TSSs, respectively" (p.6, l.143).

- The last part about the crossover TSS event is the most interesting (p.6) but it is unfortunately under-developed. How are these events detected? I could not find any detail about this method. Is it specific? What is the exact "switching" criteria that is measured? Again, what would be the expected distribution of that criteria when comparing called TSSs between biological replicates? Regarding the results, are the proportions given in p.6 l.150 relative to the total number of genes? What would that be relatively to the number of tested genes, i.e. with multiple TSSs in both species? It would be interesting to see these results along with the source data -read coverage, TSS position, normalized expression values- for one or several examples in a genome browser. Such an illustration could bring some value to the study.

- The HSP analysis would need a negative control or an estimation of a baseline level, using the average number of TSSs in the other genes for instance, or in genes with similar levels of expression. The authors mentioned a statistical test (p.7 l.172) but what was tested exactly is not clear ("more variation was seen": how was this variation measured, what are the numbers, what is the null hypothesis/model?). It might be that the observed difference in number of TSSs is simply due to a higher expression level (see major issues above). A t-test is mentioned again (p.8, l.184) with insufficient information. Differential expression analysis requires dedicated tools like edgeR or DESeq2, although the low number of replicates might not allow to perform any in this study.

- I do not see the claimed "association between TSS expression and peak shape in different genes" in Figure 4 (p.7, l.188).

- Comparison between developmental stages: "In the current study, about 8.7% ... expressed genes showed divergent single/multiple TSSs usage". The definition of "divergent single/multiple TSSs usage" is missing. The precise measure has to be detailed. No significance is estimated.

- "About 50-64% of genes with a single TSS expressed in fetal liver, showed an increase in the number of TSS with aging from single to multiple.". What are the numbers? Is this increase significant?

- p.11, l.290: "Interestingly, a noticeable proportion of the genes (40.1%) had divergent TSS numbers". Isn't that just expected? What is the same proportion between replicates from the same

tissue?

- Figure 6: the method is not presented and details are missing: the correlation method (Pearson, Spearman, etc), the distance metric, the software/tool for the hierarchical clustering.

C. Minor issues

Please refer to Table 1 at the very beginning of the Results, where details such as number of replicates are mentioned. It would make it easier for the readers to quickly get a clear idea of the complete experimental design.

Table 1 is informative but it is difficult to visualize the correlation between the number of CAGE tags and the number of called TSSs. Adding a graph would help to provide a broad picture to the readers. It would also give the opportunity to include (some of?) the results of the subsampling analysis from Supplementary Table 1.

In Table 1, numbers are difficult to read and compare: please insert comas or at least right-align them instead of centering. Please also indicate the corresponding t_i threshold that has been used to call TSSs for each sample.

Methods: What is the reference annotation exactly?

p.8 l.206: "In the current study, only TSS located in the promoter, proximal, and 5' UTR regions of known genes were used for all analyses." It seems a pity to discard so much valuable data. How much is that? What is considered "proximal"? Methods must be detailed enough for the analysis to be reproduced.

p.6 l.157: "would have to [be] confirmed by"

p.7 l.163: "With the hypothesis that heat shock proteins may be involved in this adaptation". Isn't this hypothesis supported by existing literature in other species?

l.166: Why would the two genes BAX and BCL2 be included in the HSP analysis?

p.8 l.205: "A small proportion of TSSs observed within introns could be the result of recapping due to post-transcriptional modifications." I think that recapping was reported in exons, not in introns.

p. 11, l.284: "In general, 4% (thyroid) to 16% (lung) of total TSS peaks identified in known gene transcripts overlapped with coding sequence (CDS), intron, exon, 3' UTR , and antisense (i.e. genes on the opposite strand) regions."

A figure or table would help. This could be done for both subspecies.

Figure 9: "Interactive graph": this is not an interactive graph.

Figure legends: in general, the legend should first describe the content of the represented information, not the form. Descriptions like "Histogram of" or "Heat map of" could be removed. For instance, "Fig.7. Frequency histogram of the number of TSS per gene..." could be replaced by "Fig.7. Number of TSS per gene..."

The version of all used software has to be precised (including CAGEfightR, DAVID, REVIGO)

References: formatting issue in several authors lists. Journal name in author list in ref 15 and 16 for instance.

Reviewer #2 (Remarks to the Author):

This manuscript uses CAGE sequencing to profile genome wide TSS in *Bos taurus taurus* and *Bos taurus indicus* cattle to look at evolutionary divergent TSS across the two sub-species and the underlying relevance to adaptation. The authors found differential TSS usage across tissues in both sub-species and observed that the number of TSS in heat shock genes varied to a greater extent in *indicus* relative to *taurus* cattle. Their results indicate that TSS usage could be driving breed formation and sub-species divergence.

To my knowledge this is the first time CAGE sequencing data has been used in a livestock species to investigate the underlying biology of complex traits and as such this manuscript provides an excellent demonstration of the utility of 5' sequencing data beyond functional annotation.

Two recent manuscripts using 5' sequencing technologies for functional annotation in cattle (<https://doi.org/10.1101/2020.09.05.284547>) and sheep (<https://doi.org/10.1101/2020.07.06.189480>) are either under review or in press. As such this manuscript is particularly exciting and timely, increasing the complexity of the data sets available, and profiling TSS in two sub-species and in more than one developmental stage, to gain new insights into the evolution of TSS.

The only caveat I can see with the study is that only one biological replicate is included for the majority of tissues analysed in the study. This is often the case for studies involving large mammals, for example, the initial goals of the FAANG consortium were to annotate the genomes of single reference animals per species to a high resolution. I don't think the number of biological replicates affects the novelty, interest and importance of the work. The sheep and cattle papers mentioned above included only one and four biological replicates, respectively. It is a limitation of the study that ought to be indicated somewhere in the manuscript though. I think it would be sufficient to acknowledge this in the conclusions indicating that analysis of additional animals (and potentially additional tissues relevant to divergent traits between the two subspecies) would increase the resolution of the findings, particularly in the comparison of expressed TSS between the two sub-species.

The methods used to account for the lack of biological replicates and ensure the findings are statistically robust are clearly demonstrated and described.

Specific Line Changes

Main manuscript

Line 40 'Closely separated' is a bit counter intuitive, I'm not sure what terminology would be better though, maybe 'adjacent' or change to 'several TSS located in close proximity'?

Line 59 I'm not sure 'numerous' is necessary here if only one study is referenced?

Line 143 italics '*indicus*'

Line 142-145 This sentence is long and quite complicated. Could it be split into two? Similarly for lines 136-140.

Line 145 'Left panel' is a bit ambiguous, could you label these? (see also comment on the figure below).

Line 163-167 Again this is quite a long sentence that could be split into two.

Line 181 'Greater' might be better than 'higher' here?

Line 195-196 I'm not sure the second part of this sentence is necessary, it could end at 'developmental-stage-specific genes' I think or change to 'and facilitate our understanding of developmental-stage-specific gene regulation'.

Line 323 Please include the genes in parentheses here, to help the reader follow the narrative.

Line 324 I think 'consistent with previous research' requires more information here as it's not clear whether research focused on biological function or number of TSS is being referred to?

Line 334-340 I'm not sure that this section fits very well with the rest of the narrative. Duplication events aren't described in the rest of the narrative or mentioned in the rationale for the study in the introduction. Probably starting the paragraph with 'In conclusion' is a bit misleading too as it appears to be summing up the findings of the entire manuscript which actually follows in the conclusions section. In my opinion this paragraph could be removed but the authors might prefer to reword it.

Line 345-346 Commas are missing before 'diverged' and after '(Bos primigenius)'.

Line 347 'age' should be 'developmental stage' for consistency.

Figure 2 left and right are difficult to follow could these be changed to i), ii) etc for example?

Figure 6 I'm not sure what is being shown on each axis of this figure?

In Table 1 how does the number of CAGE tags relate to the overall number of mapped reads per tissue? How did the observed number match the expected and was this consistent across tissues?

Supplementary Note

Line 42 'Single-read' should be 'single-end'.

Line 45 As the location of where the sequencing is performed is included, should where the libraries were prepared also be included?

Line 47 How were multi-mapped reads dealt with?

Reviewer Comments, Author Responses and Manuscript Changes

Referee #1: Livestock bioinformatics

This manuscript presents an analysis of CAGE sequencing data that were obtained across a large set of tissues in two cattle subspecies: *Bos taurus* and *Bos indicus*.

The analysis starts with typical steps like read mapping and peak calling, which generates a set of predicted TSSs. After several filtering steps, the number of TSSs per gene is computed and compared between subspecies. Shapes of the TSSs are also considered. Expression levels are assigned to TSSs with a focus on Heat Shock Protein genes (HSPs) and a hierarchical clustering of tissues is realized based on these values. TSS numbers per gene are again compared, this time between adult and fetal tissues, and used to correlate tissues.

Overall, many comparisons and observations are made using this data but most of them lack a proper assessment of statistical significance. This is mostly due to an incompatibility between the experimental design and the analysis approach. This is a strong limitation to draw reliable conclusions from the results. I believe that the analysis could greatly be improved, in particular by changing the way replicates are taken into account. At several occurrences the methods should be clarified, better documented and cleared of potential biases. The results section is a bit repetitive and would benefit from more biological relevance and novelty. The CAGE sequencing data itself is of interest to the community though.

(A) Major general issues

Comment A 1: Lack of significance: Due to the lack of replicates, the experimental design does not allow to support the conclusions from most of the comparative analyses. For instance, since only one adult cow and one fetus are available in the *B. indicus* group, the comparison with the *B. taurus* group is strongly limited, in particular with such a little set of common tissues. Consequently, the reported differences cannot be reliably assigned to the factor of interest (the subspecies). Same goes for the adult vs. fetus comparison, with only one and two fetuses from the *indicus* and *taurus* subspecies respectively.

I would suggest the authors to strengthen their experimental design, adjust their claims, or change the scope of the analysis. Instead of using the replicates to discard an unreported proportion of the called TSSs, one way to proceed could be for instance to estimate for each tissue the distance distribution between proximal TSSs from the available duplicates in the

B. taurus group. This could give an estimation of the expected difference to be observed between samples from the same subspecies (cf. Fig.1), and potentially help to compare samples from the indicus animal. The liver tissue is actually the only one with biological replicates in all the groups (one adult and one fetus from indicus, one fetus and two adults from taurus).

An alternative way to design the analysis would be to focus on the multiplicity of available tissues to better characterize the diversity of TSS usage and explore regulatory aspects through a rigorous analysis of a tissue-specificity criteria for instance. This would better fit the available data than a pairwise-oriented comparative approach between groups with insufficient numbers of replicates.

Response: Thank you. We found your comments extremely helpful and have revised accordingly. To study the evolutionary TSS, pairwise comparisons between *Bos taurus* and *Bos indicus* sub-species for spleen, muscle and adult liver were carried out by following the workflow in Fig. 1. To strengthen our experimental design, instead of using the replicates to discard an unreported proportion of the called TSSs, biological replicates after doing quality control and selecting the CAGE tag starting sites supported by 3 or more CAGE reads in a single sample were merged together. Therefore, the resulting library contained a union of CTSS present in the *Bos taurus* biological replicates and raw tag counts for those CTSS in available replicates (see Methods, “scenario1”). Besides, to support our conclusion from comparative analysis between sub-species and assigning the reported differences to the sub-species effect, as you suggested we performed second scenario (see Methods, “scenario2”), investigating consensus TSS cluster properties including the distance distribution between dominant TSSs (see “Supplementary Fig. 2-4C”) for each tissue from the available duplicates in the *B. taurus* sub-species. Moreover, to focus on the multiplicity of available tissues to better characterize the diversity of TSS usage and explore regulatory aspects, TSSs were called for each tissue and then to be able to compare tissues at the level of clusters of TSSs, TSS clusters from all tissues were aggregated into a single set of consensus TSS clusters (Fig. 1) separately for each sub-species. Then, for each sub-species the distance distribution between

dominant TSSs, proportion of TSS clusters with differential TSS usage and significant 'shifting' TSS clusters for pair of tissues were estimated (see Methods, "scenario5").

Comment A 2. Issue of the "TSS number per gene" metric.

A large part of the results are based on comparing the number of TSS per gene between groups of samples. This number depends on the total number of predicted TSS, which is explicitly adjusted by the authors for each tissue during the peak calling step (with the threshold parameter t_i).

First, it does not seem relevant to me to normalize read coverage by adjusting the number of predictions, in particular when it is done by raising a threshold detection in abundant datasets. This might lead to a drastic and unnecessary loss of data by artificially reducing the number of detected TSSs in large libraries. There is nothing wrong in obtaining more TSSs when sequencing deeper, it is a natural way to detect transcripts with low expression levels. So as long as noise saturation is not reached and called peaks indicate genuine TSSs (which is not not addressed here), there is no reason to remove valid predictions using the worse sample as a reference. Transcriptome studies based on RNA-seq do not reduce the number of predicted genes/transcripts to compare samples but use instead proper coverage normalization methods. In addition, removing TSSs that do not appear at the exact same position in several replicates as indicated in the Sup. Methods (p.4, l.85) is certainly excessive. The CAGEfightR manual explicitly recommends to use the "support" threshold for cases with "a very large number of and/or very noisy samples (For example due to poor RNA quality)", which does not seem to be the case here.

Second, the number of TSS per gene depends on several confounding factors that are not controlled, including the amount of sequencing data (number of tags) and the expression level of the genes. It might be for instance that an observed difference in number of TSSs is uniquely due to a global increase of gene expression level, which allows for a more comprehensive profiling of the transcriptome complexity while no active biological regulation could actually be involved at the level of TSS usage. These confounding factors need to be properly investigated and addressed before considering the number of TSS per gene as a reliable metric.

Response: Thanks for your comments. The more systematic investigation of several CAGE dataset has shown that the reverse cumulative distribution of the number of tags per TSS follow a power-law distribution to a really good approximation (Balwierz et al., Genome Biology 2009). Therefore, we did normalization using the power-law method implemented in CAGEr in

the revised manuscript. Furthermore in agreement with your comments given that the number of TSSs can vary according to different factors such as the method of TSSs calling and also sample coverage we decided to change the scope of the study from comparison of TSSs number per gene to investigation of the TSS cluster properties across the tissues/replicates.

Comment A 3. Issue of the GO analysis

A Gene Ontology (GO) term enrichment analysis needs a carefully defined gene set to be provided as a reference background. When no relevant gene set of reference is available, it is possible to provide the computational tool with the entire set of known genes for that species. This is what has been done in that study (p.4, l.91 of the Sup Methods: "Bos taurus genome was considered as background in all analyses"). In this case however, as in many transcriptome studies, the reference should be defined as the set of the expressed genes, not the set of all known genes. For instance, when looking for GO term enrichment in a subset of differentially expressed genes, the reference background needs to be the set of the expressed genes, including the differentially and the non-differentially expressed ones, but excluding non-expressed genes that were not tested for differential expression. Similarly, in this study, when performing a GO term enrichment analysis on a subset of genes, for instance expressed genes "with a single TSS" (p.8 l.213), the background must be limited to the set of all expressed genes, including the ones with a single TSS and the one with several TSSs, but not the ones with no expression and consequently no predicted TSS. Otherwise it is not possible to define if the GO term enrichment is due to the factor of interest (the specific subset) or to the simple fact that the subset of genes belongs to the larger set of expressed genes, and any random subset could lead to the same enrichment. All GO analyses should be fixed accordingly.

Response: All GO analyses were fixed according to the reviewer's suggestion and for each pairwise comparison the reference was defined as the set of the all expressed genes for that analysis.

Comment A 4. Gene expression analysis

Gene expression analyses are incorrect or insufficiently documented. For instance p.7, l.184 ("t-Test") or p.10, l.252. ("We performed quantitative and expression analysis of CAGE-Seq data", with no further detail). It should be noted that the CAGEfightR package used in this study generates output tables that are compatible with proper gene expression analysis tools like edgeR or DESeq2.

Response: The reviewer is correct, we didn't do any expression analysis as we have only one replicates for *indicus* tissues. The sentences suggesting we did in the manuscript have been removed.

Comment A 5. Novelty and biological significance of the results

Differences are reported between samples and some variation is observed across tissues, but all this seems quite expected to me and do not bring new conclusions considering existing reports on human/mouse. The data is original and has potential, but no strong biological results is clearly drawn from the study.

The comparison with a reference annotation like REFSEQ or ENSEMBL is curiously missing. The distribution of the predicted TSSs in the diverse types of genomic regions should be provided, including the proportion of TSSs in promoters, near an annotated TSS, in 5'UTR, etc. Also, the distance distribution of the predicted TSSs with respect to the annotated ones would greatly help to show the value of the data.

Response: We have now annotated the CAGE-seq dataset with the ENSEMBL *Bos taurus* ARS-UCD1.2 annotation (gene, transcript, and exon coordinates) and the graph showing the proportion of CTSSs in different genomic regions (intron, exon, promoter and intergenic regions) is provided in Supplementary Fig. 8. Moreover, the distance distribution of the predicted TSSs with respect to the annotated ones (Ensemble release 102 and assembly release 106) for *Bos indicus* adult liver tissue was compared (p.13, l.313). Also, for all genes reported in figures, predicted TSS along with annotated ones are shown as an example (Figure 2-4D and 6; Supplementary Fig. 2-4D and 6-7). Moreover, the 1000 bull genomes dataset (run8) was investigated to find the SNPs with significant shift in allele frequency among significant differentially used TSS regions across sub-species based on the F_{ST} method.

Comment A 6. CAGEfightR was used to call the TSSs but apparently not to detect enhancer RNAs (using the bidirectional clustering function of the package). Is there a reason for that? Considering the broad set of tissues in this study, identifying enhancers and, if possible, the tissue-specific or ubiquitous ones, might provide valuable results and biological insights.

Response: We are currently investigating the enhancers but the results will be published in a new manuscript as there are a lots of biological results which is hard to put in to only one paper.

Comment A 7. Several steps of the analysis are not clear or insufficiently documented.

Several statistical tests are performed without providing enough details about the size of the

groups, the exact numbers or proportions that are compared, etc. The writing should be improved to better explain the methods (see detailed comments below).

Response: More clarification has been added, the methods section is now better documented.

Comment A 8. Last but not least: data availability. Sequencing data must be available to allow publication. Sequences have to be submitted to a dedicated public database (SRA, ENA) and the accession codes provided in the manuscript. The data availability statement "For data availability please contact the authors." in the Reporting Summary is not acceptable.

Response: We will submit the bam files for all tested samples at FAANG.

B. Detailed comments

Comment B1. About the "statistical analysis of the reproducibility" (page 3, bottom).

- From Table 1 and S1, it seems that there is a positive correlation between the number of CAGE tags and the number of called TSSs. This is an important point that should be addressed and commented in the text.

Response: The correlation between number of CTSS and TSS were measured and addressed in the main text (p.5 l.90).

Comment B2. The impact of the number of replicate, however, was not obvious to me at all. Since no results from TSS calling are presented on individual replicates, I first thought that replicates were simply merged together before TSS calling, which is an acceptable strategy once replicates are shown to be consistent. The only clue I could find about the process was one brief mention in the supplementary note: "TSSs appearing in only a single replicate were discarded" (p.4, l.85). Does it mean that only TSSs found exactly at the same position in different replicates were kept? This would imply that two adjacent TSSs for instance would be both discarded and none of them would be considered. Is that correct? This seems like a very stringent and restrictive filter to me: how many TSSs have been discarded like this?

Response: In the revised manuscript, biological replicates were merged together where applicable (see Methods, scenario1). Therefore, the resulting library contained a union of

CTSS present in the *Bos taurus* biological replicates and raw tag counts for those CTSS in available replicates. As described above, the analysis has now been much more clearly documented, we hope.

Comment B3. If replicates have been processed separately please clarify in the text and provide the raw results in Table 1 (the number of called TSSs for instance are missing for replicates 2 and 3) and in Supplementary Table 1 as well (number of TSSs in each biological replicate and subsamples). Please also clarify the merging process of the results from the replicates and the subsequent outcomes, in particular when 3 replicates are available.

Response: The number of consensus TSS are shown in supplementary table 1-3 and table 1-2. The merging process was explained in detailed in response to comment A1 and in Methods, "*scenario1*".

Comment B4. "for samples with high coverage [...], to avoid missing a noticeable number of true TSSs, the t_i was set to 25 tag counts" (p.4 l.87): this sounds strange to me because missing true positive is more likely in samples with low coverage, not high coverage.

Response: In the revised version the power law method implemented in CAGEr was used for normalization. Refer to response to comment A2 for more detailed information and Methods, "*Tag count Normalization*".

Comment B5. Assessing the reproducibility based on the sole number of predicted TSSs is not satisfying. The proportion of identical/proximal predictions would be more informative. Consider a Venn diagram for instance. If only the number of predictions is considered and not their nature, the statement of 80% reproducibility has to be removed (p.4, l.94). Showing a strong correlation/consistency between replicates would support the general reliability of the data. Idem for subsampling results: how close are TSSs predicted from subsamples?

Response: The reproducibility was measured as the fraction of consensus TSS clusters commonly observed in the total and half samples. Also, to investigate the effect of coverage on TSS diversity of each consensus TSS cluster we measured Pearson correlation between Shannon index of TSS diversity of consensus TSS cluster in sub-sample and total samples (Supplementary Fig. 1).

Comment B6. The part on classifying TSSs into shape categories (p.5 l.119) is not clear and needs to be rewritten.

- A detailed presentation of an existing classification is first made. Then it is difficult to

understand why the authors do not follow that classification. The sentence "[...] but in the current study to better visualize the probability distribution of the cluster of tags within a TSS we classified TSS into four shape categories based on the genomic interval covered by the cluster of tags and the probability of the tags distribution within a single TSS" is not consistent with the proposed classification. "In the single dominant peak class (SP), the tags were concentrated to no more than four consecutive start positions." Is that an observation using the previous definition of the SP class or is it a new definition of the class? How is that observed exactly or applied in practice? If it is a new criteria to define the SP class, why is the former criteria of the IQR size used in the next sentence ("The clusters spanning a broader region (IQR>4 bp)")?

Response: Shape classification of TSS shape for HSP genes was done based on the study of Carninci, P. *et al.* *Nature Genetics* (2006) 38:626-635, and a sentence has been added to clarify this. The misleading sentences have been removed in the text.

Comment B7. Figure 3: the proportion of the TSS in each category is then compared between subspecies within groups of TSSs. No statistical significance is proposed to support the potential relevance of these observations. According to the figure, the TSS distribution within categories seems highly variable, so even comparing random subsamples might lead to very different proportions. Biological relevance is also missing from these observations. For instance I do not see what conclusion could be drawn from an observation like "TSSs in indicus tended to be wider, spread over a larger genomic region, in genes with divergent TSSs in which indicus and taurus sub-species had single and multiple TSSs, respectively" (p.6, l.143).

Response: To address existed criticisms we had to substantially revised manuscript. Therefore all of the figures and analysis was changed by the changes made in the scope of the analysis from number of TSSs to discovery of TSSs properties in each tissues/replicates, as suggested in the major comments. To support the potential relevance of most of the observation, bootstrapping method was used and significance level was reported for each statistic tests.

Comment B8. The last part about the crossover TSS event is the most interesting (p.6) but it is unfortunately under-developed. How are these events detected? I could not find any detail about this method. Is it specific? What is the exact "switching" criteria that is measured? Again, what would be the expected distribution of that criteria when comparing called TSSs between biological replicates? Regarding the results, are the proportions given

in p.6 l.150 relative to the total number of genes? What would that be relatively to the number of tested genes, i.e. with multiple TSSs in both species? It would be interesting to see these results along with the source data -read coverage, TSS position, normalized expression values- for one or several examples in a genome browser. Such an illustration could bring some value to the study.

Response: Crossover TSS was only investigated for the *Bos taurus* fetal and adult liver samples where there were two biological replicates for each developmental stage. The Methodology for crossover switching events was explained in more detail in Material and Methods (see Methods, "scenario 4"). The list of the genes having cross over TSSs was provided in Supplementary Table 11.

Comment B9. The HSP analysis would need a negative control or an estimation of a baseline level, using the average number of TSSs in the other genes for instance, or in genes with similar levels of expression. The authors mentioned a statistical test (p.7 l.172) but what was tested exactly is not clear ("more variation was seen": how was this variation measured, what are the numbers, what is the null hypothesis/model?). It might be that the observed difference in number of TSSs is simply due to a higher expression level (see major issues above). A t-test is mentioned again (p.8, l.184) with insufficient information. Differential expression analysis requires dedicated tools like edgeR or DESeq2, although the low number of replicates might not allow to perform any in this study.

Response: The assessment of differential TSS usage across sub-species for HSP genes was obtained by performing a two sample Kolmogorov-Smirnov test on cumulative sums of CAGE signal along the consensus TSS to assess whether the two underlying probability distributions differ across sub-species for spleen, liver, muscle. Because of the low number of replicates we could not perform any gene expression analysis, we only provided the heat map of the TSS expression for each gene across different tissues (Fig.5).

Comment B10. I do not see the claimed "association between TSS expression and peak shape in different genes" in Figure 4 (p.7, l.188).

Response: The misleading sentences have been removed in the text.

Comment B11. Comparison between developmental stages: "In the current study, about 8.7% ... expressed genes showed divergent single/multiple TSSs usage". The definition of "divergent single/multiple TSSs usage" is missing. The precise measure has to be detailed. No significance is estimated.

"About 50-64% of genes with a single TSS expressed in fetal liver, showed an increase in the number of TSS with aging from single to multiple.". What are the numbers? Is this increase significant?

p.11, l.290: "Interestingly, a noticeable proportion of the genes (40.1%) had divergent TSS numbers". Isn't that just expected? What is the same proportion between replicates from the same tissue?

Response: All of the figures and analysis related to comparison of TSSs number across tissues/sub-species/stages were removed as the scope of the analysis was changed from comparison of TSSs number to discovery of TSSs properties in each tissues/replicates. The proportion of divergent TSSs (TSSs only observed in one replicates) between biological replicates from the same tissue was shown in main text (p.6 l.118) and also (Supplementary) Fig 2-4A. Also significant estimation based on the bootstrap method is reported for all observed statistics in main text.

Comment B12. Figure 6: the method is not presented and details are missing: the correlation method (Pearson, Spearman, etc), the distance metric, the software/tool for the hierarchical clustering.

Response: Now addressed in the Methods section.

C. Minor issues

Comment C1. Please refer to Table 1 at the very beginning of the Results, where details such as number of replicates are mentioned. It would make it easier for the readers to quickly get a clear idea of the complete experimental design.

Table 1 is informative but it is difficult to visualize the correlation between the number of CAGE tags and the number of called TSSs. Adding a graph would help to provide a broad picture to the readers. It would also give the opportunity to include (some of?) the results of the subsampling analysis from Supplementary Table 1.

In Table 1, numbers are difficult to read and compare: please insert comas or at least right-align them instead of centering. Please also indicate the corresponding t_i threshold that has been used to call TSSs for each sample.

Response: Now fixed. Also, the correlation between number of CTSS and TSS were measured and addressed in the main text (p.5 l.90).

Comment C2. Methods: What is the reference annotation exactly?

p.8 l.206: "In the current study, only TSS located in the promoter, proximal, and 5' UTR regions of known genes were used for all analyses." It seems a pity to discard so much valuable data. How much is that? What is considered "proximal"? Methods must be detailed enough for the analysis to be reproduced.

Response: CAGE-seq dataset has been annotated with ENSEMBL Bos taurus ARS-UCD1.2 annotation (gene, transcript, and exon coordinates) and the graph showing the proportion of TSSs in different genomic regions is provided in Supplementary Fig. 8.

Comment C3. p.6 l.157: "would have to [be] confirmed by"

Response: It was fixed.

Comment C4. p.7 l.163: "With the hypothesis that heat shock proteins may be involved in this adaptation". Isn't this hypothesis supported by existing literature in other species? l.166: Why would the two genes BAX and BCL2 be included in the HSP analysis?

Response: A previous study (Hooper HB, Dos Santos Silva P, de Oliveira SA, Merighe GKF, Negrão JA. Acute heat stress induces changes in physiological and cellular responses in Saanen goats. Int J Biometeorol. 2018 Dec; 62(12):2257-2265. doi: 10.1007/s00484-018-1630-3. Epub 2018 Oct 27. PMID: 30368674.) revealed that the acute heat stress could increase the expression of heat shock proteins (HSP60, 70, and 90) and genes related to apoptosis (e.g. BAX, Bcl-2), suggesting that these genes have protective functions

Comment C5. p.8 l.205: "A small proportion of TSSs observed within introns could be the result of recapping due to post-transcriptional modifications." I think that recapping was reported in exons, not in introns.

Response: Now fixed (p.11 l.245).

Comment C6. p. 11, l.284: "In general, 4% (thyroid) to 16% (lung) of total TSS peaks identified in known gene transcripts overlapped with coding sequence (CDS), intron, exon, 3' UTR , and antisense (i.e. genes on the opposite strand) regions." A figure or table would help. This could be done for both subspecies.

Response: It was shown in supplementary Fig. 8.

Comment C7. Figure 9: "Interactive graph": this is not an interactive graph. Figure legends: in general, the legend should first describe the content of the represented information, not the form. Descriptions like "Histogram of" or "Heat map of" could be removed.

For instance, "Fig.7. Frequency histogram of the number of TSS per gene..." could be replaced by "Fig.7. Number of TSS per gene...".

The version of all used software has to be precise (including CAGEfightR, DAVID, REVIGO)

Response: Now fixed in Methods.

Comment C8. References: formatting issue in several authors lists. Journal name in author list in ref 15 and 16 for instance.

Response: Now fixed.

Referee #2: Livestock genomics and transcriptomics

Reviewer #2 (Remarks to the Author):

This manuscript uses CAGE sequencing to profile genome wide TSS in *Bos taurus taurus* and *Bos taurus indicus* cattle to look at evolutionary divergent TSS across the two sub-species and the underlying relevance to adaptation. The authors found differential TSS usage across tissues in both sub-species and observed that the number of TSS in heat shock genes varied to a greater extent in *indicus* relative to *taurus* cattle. Their results indicate that TSS usage could be driving breed formation and sub-species divergence.

To my knowledge this is the first time CAGE sequencing data has been used in a livestock species to investigate the underlying biology of complex traits and as such this manuscript provides an excellent demonstration of the utility of 5' sequencing data beyond functional annotation.

Two recent manuscripts using 5' sequencing technologies for functional annotation in cattle (<https://doi.org/10.1101/2020.09.05.284547>) and sheep (<https://doi.org/10.1101/2020.07.06.189480>) are either under review or in press. As such this manuscript is particularly exciting and timely, increasing the complexity of the data sets available, and profiling TSS in two sub-species and in more than one developmental stage, to gain new insights into the evolution of TSS.

Comment 1. The only caveat I can see with the study is that only one biological replicate is included for the majority of tissues analysed in the study. This is often the case for studies involving large mammals, for example, the initial goals of the FAANG consortium were to annotate the genomes of single reference animals per species to a high resolution. I don't think the number of biological replicates affects the novelty, interest and importance of the work. The sheep and cattle papers mentioned above included only one and four biological replicates, respectively. It is a limitation of the study that ought to be indicated somewhere in the manuscript though. I think it would be sufficient to acknowledge this in the conclusions indicating that analysis of additional animals (and potentially additional tissues relevant to divergent traits between the two subspecies) would increase the resolution of the findings, particularly in the comparison of expressed TSS between the two sub-species.

The methods used to account for the lack of biological replicates and ensure the findings are statistically robust are clearly demonstrated and described.

Response: Thank you for your positive feedback. To account for the lack of biological replicates and ensure the findings are statistically robust and support our conclusion from comparative analysis between sub-species, the pairwise comparisons using the same

workflow were applied between the *taurus* biological replicates separately for spleen, muscle and liver (Methods, “*scenario1 and2*”). This resulted in an estimation of the expected differences to be observed between biological samples from the same sub-species. Also, the suggested manuscript/papers were cited in the main text (p.4 l.67). Moreover, we highlighted that analysis of additional animals (and potentially additional tissues relevant to divergent traits between the two subspecies) would increase the resolution of the findings, particularly in the comparison of expressed TSS between the two sub-species (p.9 l.208 and in Conclusions).

Comment 2. Specific Line Changes

Main manuscript

Line 40 ‘Closely separated’ is a bit counter intuitive, I’m not sure what terminology would be better though, maybe ‘adjacent’ or change to ‘several TSS located in close proximity’?

Response: Was changed to ‘adjacent’ (p.3. l.44).

Line 59 I’m not sure ‘numerous’ is necessary here if only one study is referenced?

Response: Was changed to ‘some’ (p.4. l.74).

Line 143 italics ‘indicus’

Response: Was fixed.

Line 142-145 This sentence is long and quite complicated. Could it be split into two? Similarly for lines 136-140.

Response: Both sentences were removed as the scope of analysis was changed.

Line 145 ‘Left panel’ is a bit ambiguous, could you label these? (see also comment on the figure below).

Line 163-167 Again this is quite a long sentence that could be split into two.

Line 181 ‘Greater’ might be better than ‘higher’ here?

Response: Was fixed.

Line 195-196 I’m not sure the second part of this sentence is necessary, it could end at ‘developmental-stage-specific genes’ I think or change to ‘and facilitate our understanding of developmental-stage-specific gene regulation’.

Response: It was ended at ‘developmental-stage-specific genes’ (p.10 l.237).

Line 323 Please include the genes in parentheses here, to help the reader follow the narrative.

Line 324 I think 'consistent with previous research' requires more information here as it's not clear whether research focused on biological function or number of TSS is being referred to?

Response: The total paragraph was removed as the scope of analysis was changed.

Line 334-340 I'm not sure that this section fits very well with the rest of the narrative. Duplication events aren't described in the rest of the narrative or mentioned in the rationale for the study in the introduction. Probably starting the paragraph with 'In conclusion' is a bit misleading too as it appears to be summing up the findings of the entire manuscript which actually follows in the conclusions section. In my opinion this paragraph could be removed but the authors might prefer to reword it.

Response: 'In conclusion' was removed from the beginning of the sentence (p.15 l.362).

Line 345-346 Commas are missing before 'diverged' and after '(Bos primigenius)'.

Line 347 'age' should be 'developmental stage' for consistency.

Figure 2 left and right are difficult to follow could these be changed to i), ii) etc for example?

Figure 6 I'm not sure what is being shown on each axis of this figure?

Response: All of the specific notes have been fixed.

In Table 1 how does the number of CAGE tags relate to the overall number of mapped reads per tissue? How did the observed number match the expected and was this consistent across tissues?

Response: Correlation between CAGE tags and mapped reads was calculated and addressed in the main text (p.5 l.90).

Supplementary Note

Line 42 'Single-read' should be 'single-end'.

Response: It was fixed (p.17 l.424).

Line 45 As the location of where the sequencing is performed is included, should where the libraries were prepared also be included?

Response: It was included (p.17 l.426).

Line 47 How were multi-mapped reads dealt with?

Response: Only primary alignments with a quality of greater than 20 (<99% chance of true)

were considered (p.19 l.460).

REVIEWERS' COMMENTS:

Reviewer #1 (Remarks to the Author):

The analysis and the manuscript have been drastically changed and improved. In particular, the comparative analysis between subspecies in the three tissues, along with its internal control between biological replicates of the same subspecies, is clean and very relevant. Results are impressive and convincing. The comparative analysis between developmental stages has been largely modified too, and the parallel processing of the three datasets (taurus liver, indicus liver and indicus lung) gives enough context to put the results in perspective. Missing details about the methods (parameters, software versions, etc) have been provided.

Globally, considerable efforts have been made to address the issues raised in the original review, resulting in a different and much better manuscript. Congratulations to the authors for this work.

The only remaining modifications I would suggest are (see below for details):

- 1) A few words should be changed p.5 about the "reproducibility" analysis.
- 2) The penultimate paragraph about the "error hypothesis" could optionally be dropped.
- 3) Sequencing data must be provided (accession IDs to SRA/ENA) along with the corresponding metadata.

Details.

l.97-112: there is a confusion about the term "reproducibility".

Reproducibility is usually assessed between biological replicates, either from different organisms or at least different samples from the same tissue. Here the authors investigate some kind of "technical reproducibility" instead, within the same sample. What is performed is more a saturation analysis of each sequencing library than a "reproducibility" analysis. It is informative, because a good correlation between subsets means that the library has been sequenced deep enough, but it does not allow to conclude anything about reproducibility across various libraries. So, statements about reproducibility should be removed, in particular the one at l.112: "so reproducibility should be high".

Actually, according to the numbers of CTSS in individual replicates vs. all replicates (Sup. Table 1), it looks like a substantial proportion of the CTSS from different replicates do not overlap. I would say that maybe 30-50% of the CTSS from each sample are specific to that replicate, so contrary to what is claimed reproducibility does not seem high to me. This is further supported by results from the GO analysis between biological replicates (Sup Table 6).

The comparative analysis between subspecies was carried in three tissues separately, by identifying among the consensus TSS clusters between taurus and indicus (above an expression threshold) the ones with a differential usage/shifting score. To give an idea of the expected variability, the same comparative analysis was performed in each three tissues between two biological replicates of the same subspecies (taurus). The difference between the proportion of significant TSSs in Table 1 vs. Sup. Table 3 is very convincing. In particular, the results of the liver TSSs are spectacular. The fact that the GO analysis was carried on the control too (between biological replicates) is also very informative, although it is a bit surprising that no difference was found in liver. Just to make sure: any missing results in Sup Table 6?

A inherent yet minor limitation of this analysis is that all the CAGE data analysis has been performed on the only available genomic sequence of the *Bos taurus* subspecies. The fact that reads from indicus were mapped and processed on a reference genome from the other subspecies could be responsible for part of the differences that have been observed. However, the considerable difference between the results in liver vs. the other tissues rules out the hypothesis of a major bias coming from the nature of the reference genome.

A second potential limitation is that most of the significant changes in TSS usage and shifts seem to

affect slightly the resulting mRNA in terms of size and structure, likely with no impact on the coding frame. Many cases of alternative TSS usage are reported that alter the exon-intron structure (for instance changing the first exon) and/or the encoded protein. In this study, comparative analyses of common consensus TSSs might not capture this transcriptome diversity. In this regard, the TSS crossover study provides an interesting complement.

The weakest part of the manuscript in my opinion is the "error hypothesis" investigation about the HK genes. The observation that housekeeping genes are expressed with less variability than the other genes across tissues only confirms that they might indeed be genuine housekeeping genes, nothing more. This does not bring anything new in my opinion, and does certainly not confirm an "error hypothesis". Such conclusion is almost a circular reasoning. I would not recommend to include that analysis in the final version, it would keep the study stronger and shorter.

Last but not least: I am pretty sure I already raised this point, but accession numbers of the sequencing data are still missing. l. 606: "All bam files were submitted at FAANG portal.": this is not enough. The original sequencing data must be registered to SRA or ENA and an accession number has to be provided in the manuscript before publication. This includes all fastq.gz read files along with the associated metadata (tissue, animal, subspecies, developmental stage) for the 25 samples.

Minor/typos:

l.22: "across sub-species" is repeated

Fig1 "pawer" => power

l. 238: "separately for fetal and adult samples separately"

l.253: "A Significantly" => lower case s

l. 269: "Although [...], but" => remove the but

Reviewer #2 (Remarks to the Author):

The authors have done an efficient and thorough job of addressing all the comments raised and I believe the manuscript is now suitable for publication.

I just have one small point in that the authors indicate that the data was submitted to the FAANG data portal. Submissions to the FAANG data portal are usually for the raw data via the ENA and for the sample metadata through the BioSamples database. As such there should be an accession number from the ENA and a sample accession for BioSamples. Could these please be included in the final version of the manuscript in the data availability statement. If it's just the BAM files that were submitted as analysis files via the FAANG data portal could the authors please also upload the raw data to the repositories and flag them as FAANG datasets.

REVIEWERS' COMMENTS:

Reviewer #1 (Remarks to the Author):

The analysis and the manuscript have been drastically changed and improved. In particular, the comparative analysis between subspecies in the three tissues, along with its internal control between biological replicates of the same subspecies, is clean and very relevant. Results are impressive and convincing. The comparative analysis between developmental stages has been largely modified too, and the parallel processing of the three datasets (taurus liver, indicus liver and indicus lung) gives enough context to put the results in perspective. Missing details about the methods (parameters, software versions, etc) have been provided.

Globally, considerable efforts have been made to address the issues raised in the original review, resulting in a different and much better manuscript. Congratulations to the authors for this work.

The only remaining modifications I would suggest are (see below for details):

1) A few words should be changed p.5 about the "reproducibility" analysis.

Author: Thank you. The word "technical" as advised was added before reproducibility. As suggested the sentence at l.112: "so reproducibility should be high". Was replaced with reflecting that our libraries have been sequenced deep enough.

2) The penultimate paragraph about the "error hypothesis" could optionally be dropped.

Author: The paragraph related to error hypothesis in abstract and the main text was removed.

3) Sequencing data must be provided (accession IDs to SRA/ENA) along with the corresponding metadata.

Author: The original sequencing data was registered to ENA and an accession number was provided in the manuscript.

Details.

l.97-112: there is a confusion about the term "reproducibility".

Reproducibility is usually assessed between biological replicates, either from different organisms or at least different samples from the same tissue. Here the authors investigate some kind of "technical reproducibility" instead, within the same sample. What is performed is more a saturation analysis of each sequencing library than a "reproducibility" analysis. It is informative, because a good correlation between subsets means that the library has been sequenced deep enough, but it does not allow to conclude anything about reproducibility across various libraries. So, statements about reproducibility should be removed, in particular the one at l.112: "so reproducibility should be high".

Actually, according to the numbers of CTSS in individual replicates vs. all replicates (Sup. Table 1), it looks like a substantial proportion of the CTSS from different replicates do not overlap. I would say that maybe 30-50% of the CTSS from each sample are specific to that replicate, so contrary to what is claimed reproducibility does not seem high to me. This is further supported by results from the GO analysis between biological replicates (Sup Table 6).

The comparative analysis between subspecies was carried in three tissues separately, by identifying among the consensus TSS clusters between taurus and indicus (above an expression threshold) the ones with a differential usage/shifting score. To give an idea of the expected variability, the same comparative analysis was performed in each three tissues between two biological replicates of the same subspecies (taurus). The difference between the proportion of significant TSSs in Table 1 vs. Sup. Table 3 is very convincing. In particular, the results of the liver TSSs are spectacular. The fact that the GO analysis was carried on the control too (between biological replicates) is also very informative, although it is a bit surprising that no difference was found in liver. Just to make sure: any missing results in Sup Table 6?

A inherent yet minor limitation of this analysis is that all the CAGE data analysis has been performed on the only available genomic sequence of the *Bos taurus* subspecies. The fact that reads from indicus were mapped and processed on a reference genome from the other subspecies could be responsible for part of the differences that have been observed. However, the considerable difference between the results in liver vs. the other tissues rules out the hypothesis of a major bias coming from the nature of the reference genome. A second potential limitation is that most of the significant changes in TSS usage and shifts seem to affect slightly the resulting mRNA in terms of size and structure, likely with no impact on the coding frame. Many cases of alternative TSS usage are reported that alter the exon-intron structure (for instance changing the first exon) and/or the encoded protein. In this study, comparative analyses of common consensus TSSs might not capture this transcriptome diversity. In this regard, the TSS crossover study provides an interesting complement.

The weakest part of the manuscript in my opinion is the "error hypothesis" investigation about the HK genes. The observation that housekeeping genes are expressed with less variability than the other genes across tissues only confirms that they might indeed be genuine housekeeping genes, nothing more. This does not bring anything new in my opinion, and does certainly not confirm an "error hypothesis". Such conclusion is almost a circular reasoning. I would not recommend to include that analysis in the final version, it would keep the study stronger and shorter.

Last but not least: I am pretty sure I already raised this point, but accession numbers of the sequencing data are still missing. l. 606: "All bam files were submitted at FAANG portal.": this is not enough. The original sequencing data must be registered to SRA or ENA and an accession number has to be provided in the manuscript before publication. This includes all fastq.gz read files along with the associated metadata (tissue, animal, subspecies, developmental stage) for the 25 samples.

Minor/typos:

l.22: " across sub-species " is repeated

Fig1 "pawer" => power

l. 238: "separately for fetal and adult samples separately"

l.253: "A Significantly" => lower case s
l. 269: "Although [...], but" => remove the but

Author: all was corrected.

Reviewer #2 (Remarks to the Author):

The authors have done an efficient and thorough job of addressing all the comments raised and I believe the manuscript is now suitable for publication.

I just have one small point in that the authors indicate that the data was submitted to the FAANG data portal. Submissions to the FAANG data portal are usually for the raw data via the ENA and for the sample metadata through the BioSamples database. As such there should be an accession number from the ENA and a sample accession for BioSamples. Could these please be included in the final version of the manuscript in the data availability statement. If it's just the BAM files that were submitted as analysis files via the FAANG data portal could the authors please also upload the raw data to the repositories and flag them as FAANG datasets.

Author: Thank you. The original sequencing data was registered to ENA and an accession number was provided in the manuscript.